



# Oxygen and sulfur mass-independent isotopic signatures in black crusts: the complementary negative $\Delta^{33}$S-reservoir of sulfate aerosols?

Isabelle Genot[1,2], David Au Yang[1,3], Erwan Martin[2], Pierre Cartigny[1], Erwann Legendre[2,4], Marc De Rafelis[5]

[1]Institut de physique du globe de Paris, Université de Paris, CNRS, F-75005 Paris, France.
[2]Sorbonne Université, CNRS-INSU, Institut des Sciences de la Terre de Paris, IsteP UMR7193 Paris, France.
[3]Department of Earth and Planetary Sciences, McGill University, Montréal, Canada.
[4]LATMOS-IPSL - Sorbonne Université - Université Versailles St.-Quentin, Paris, France.
[5]GET, Université Paul Sabatier, Toulouse, France.

*Correspondence to*: Isabelle Genot (genot@ipgp.fr)

**Abstract** To better understand the formation and the oxidation pathways leading to gypsum-forming "black crusts" and investigate their bearing on the whole atmospheric $SO_2$ cycle, we measured the oxygen ($\delta^{17}O$, $\delta^{18}O$ and $\Delta^{17}O$) and sulfur ($\delta^{33}S$, $\delta^{34}S$, $\delta^{36}S$, $\Delta^{33}S$ and $\Delta^{36}S$) isotopic compositions of black crust sulfates sampled on carbonate building stones along a NW-SE cross-section in the Parisian basin. The $\delta^{18}O$ and $\delta^{34}S$, ranging between 7.5 and 16.7 ± 0.5 ‰ (n = 27, 2σ) and between -2.6 and 13.9 ± 0.2 ‰ respectively, show anthropogenic $SO_2$ as the main sulfur source (from 2 to 81 %, in average ~ 30 %) with host-rock sulfates making the complement. This is supported by $\Delta^{17}O$-values (up to 2.6 ‰, in average ~ 0.86 ‰), requiring > 60 % of atmospheric sulfates in black crusts. Both negative $\Delta^{33}S$-$\Delta^{36}S$-values between -0.34 and 0.00 ± 0.01 ‰ and between -0.7 and -0.2 ± 0.2 ‰ respectively were measured in black crusts sulfates, that is typical of a magnetic isotope effect that would occur during the $SO_2$ oxidation on the building stone, leading to $^{33}S$-depletion in black crust sulfates and subsequent $^{33}S$-enrichment in residual $SO_2$. Given that sulfate aerosols have mostly $\Delta^{33}S > 0$ ‰ and no processes can yet explain this enrichment, resulting in a non-consistent S-budget, black crust sulfates could well represent the complementary negative $\Delta^{33}S$-reservoir of the sulfate aerosols solving the atmospheric $SO_2$ budget.

## 1. Introduction

The oxidation of sulfur dioxide emitted into the atmosphere (between 100 and 110 Tg($SO_2$).yr$^{-1}$, Klimont et al., 2013) results in the formation of $H_2SO_4$ that forms sulfate aerosols; having light-scattering properties they alter the radiative balance of the planet. Furthermore, they also modify the microphysical properties of clouds through the number and size of cloud condensation nucleus process (CCN; e.g. Weber et al., 2001). Although quantified with large uncertainties, the formation of these aerosols results an Earth surface cooling (Forster et al., 2007). Indeed, sulfate aerosols are responsible for a negative radiative forcing from -0.62 to -0.21 W.m$^{-2}$, in average ~ -0.41 W.m$^{-2}$, being the most influent particles that counterbalance the greenhouse gases effect (Stocker, 2014). Uncertainties regarding the formation of sulfate aerosols relate to the large variety of oxidants and conditions (e.g. pH) but in view of their major impact on climate, a more accurate understanding of the formation of these particles is necessary.



Primary sulfate aerosols consist of sulfates formed during their emission into the atmosphere (e.g. sea-salt sulfates, combustion products, volcanic sulfates) which involves therefore a local origin (Holt and Kumar, 1991).

Secondary sulfate aerosols are formed later in the atmosphere following various oxidation pathways (oxidation via OH, $O_2$ - Transition Metal Ion (TMI), $O_3$, $H_2O_2$, $NO_2$…) and relate to a distant sulfur source (Seinfeld and Pandis, 2016).

$SO_2$-oxidation via OH radical occurs in gas-phase (homogeneous oxidation) and induces the nucleation of new

aerosols particles in the atmosphere, while the oxidation reactions by $H_2O_2$, $O_3$ and $O_2$-TMI take place in aqueous-phase (heterogeneous oxidation), leading to sulfates aerosols formation on preexisting particles including cloud droplets. Therefore, $SO_2$ oxidation by these different oxidation channels results in different size and number of aerosols particles with distinct effects on radiative balance.

Stable isotope geochemistry is a central tool to both characterize sulfur sources and quantify the different

oxidants. The $\delta$ notation used here is defined as:

$\delta = [(R / R_{std}) - 1]$ with $R = {}^{18,\,17}O / {}^{16}O$ for $\delta^{18}O$ and $\delta^{17}O$ or $R = {}^{34,33,36}S/{}^{32}S$ for $\delta^{34}S$, $\delta^{33}S$ and $\delta^{36}S$

and isotope fractionation factors are expressed as follows:

${}^{18/16}\alpha_{A-B} = ({}^{18}O/{}^{16}O)_A/({}^{18}O/{}^{16}O)_B$ with A and B being two different phases.

Given that the oxidants have distinct $\delta^{18}O$ and $\Delta^{17}O$ signatures, the $SO_2$ oxidation pathways are commonly

constrained using oxygen-multi isotope ratios ($\delta^{18}O$, $\delta^{17}O$ and $\Delta^{17}O$, defined in the following section) (Alexander et al., 2012; Bindeman et al., 2007; Jenkins and Bao, 2006; Lee and Thiemens, 2001; Martin et al., 2014; Savarino et al., 2000). Sulfur isotope fractionation during $SO_2$ oxidation by OH, $O_2$-TMI, $H_2O_2$, $O_3$ (Harris et al., 2012a; Harris et al., 2012b; Harris et al., 2013a; Harris et al., 2013b) and $NO_2$ (Au Yang et al., 2018) have been determined, so additional constraints can also be brought by S-multi isotopic compositions ($\delta^{34}S$, $\delta^{33}S$, $\delta^{36}S$, $\Delta^{33}S$

and $\Delta^{36}S$). At present, it is however difficult to reach a consistent budget for $SO_2$ oxidation (chemically and isotopically). Indeed, most of sulfate aerosols have positive $\Delta^{33}S$-values (Au Yang et al., 2019; Guo et al., 2010; Han et al., 2017; Lin et al., 2018b; Romero and Thiemens, 2003; Shaheen et al., 2014), implying either a source of $SO_2$ with $\Delta^{33}S > 0$ ‰ (which has not been identified yet as all known sources have $\Delta^{33}S \sim 0$ ‰; (Alexander et al., 2012; Lin et al., 2018b)) or more likely processes forming ${}^{33}S$-enriched sulfates reservoir from initial $SO_2$

with $\Delta^{33}S = 0$ ‰, that should be balanced by a ${}^{33}S$-depleted reservoir. As none of the studied reaction process can account for positive $\Delta^{33}S$-values of sulfate aerosols (Au Yang et al., 2018; Guo et al., 2010; Han et al., 2017; Harris et al., 2013b; Lee et al., 2002; Lin et al., 2018b; Romero and Thiemens, 2003; Shaheen et al., 2014), this leads to the suggestion that some oxidants or the $SO_2$ source either have been overlooked.

Black crusts potentially represent new ways to sample the atmosphere in urban regions at relatively global scale.

They are generally formed by the sulfation of the underlying carbonate substrate resulting in a gypsum layer (Camuffo, 1995)(Fig. 1). Due to their degradation effects of monuments and buildings, in particular because the molar volume of $CaSO_4$ is larger than that of $CaCO_3$, several studies investigated sources of sulfur in black crusts, using primarily the isotopic composition of sulfur ($\delta^{34}S$) and oxygen ($\delta^{18}O$), microscopic and mineralogical aspects. Anthropogenic sulfur was found to be the major source contributing to monuments

degradation in several localities compared to marine or volcanic sulfate sources (Longinelli and Bartelloni, 1978;





Montana et al., 2012; Montana et al., 2008; Torfs et al., 1997). Intrinsic sulfates, that are plaster, mortar or oxidized pyrite (Klemm and Siedel, 2002; Kloppmann et al., 2011; Kramar et al., 2011; Vallet et al., 2006) and sulfates from aquifer rising by capillarity (Kloppmann et al., 2014) were also identified as sulfur sources in black crusts. Black crusts being sometimes the host of microbial activity (Gaylarde et al., 2007; Sáiz-Jiménez, 1995;

Scheerer et al., 2009; Schiavon, 2002; Tiano, 2002), other studies investigated the role of bacteria in gypsum formation through sulfate reduction and/or $SO_2$ oxidation (Tiano, 2002; Tiano et al., 1975). Except the work of Šrámek (1980) measuring black crust sulfates $\delta^{34}S$ that rule out the implication of micro-organisms in their formation, no further constraint has been brought so far.  Thus, black crusts were never investigated for all the oxygen ($\delta^{18}O$, $\delta^{17}O$ and therefore $\Delta^{17}O$) and sulfur ($\delta^{34}S$, $\delta^{33}S$, $\delta^{36}S$ and therefore $\Delta^{33}S$ and $\Delta^{36}S$) isotopic ratios

contained in sulfate and more specifically, in quantifying the different oxidation channels involved in the sulfate aerosols formation in the troposphere.

In this paper, we present new isotopic composition measurements of sulfate extracted from black crusts and report significant $\Delta^{17}O$, $\Delta^{33}S$, $\Delta^{36}S$ anomalies that help to discuss oxygen and sulfur isotopic variations both in

term of source effects to elucidate their origin and in term of fractionation processes leading to black crusts formation in the Paris area.

## 2.   Mass-dependent and independent fractionations

As many chemical reactions, O- and S-isotopic compositions of $SO_2$ vary during its oxidation. Most reactions are

"mass-dependent", meaning the isotopic fractionation relies on the mass differences between the isotopes; this remains valid for most unidirectional (kinetic) and/or exchange (equilibrium) reactions. In a system with at least three isotopes, mass fractionation law at equilibrium and high temperature can be derived from its partition function (Bigeleisen and Mayer, 1947; Dauphas and Schauble, 2016; Urey, 1947; Young et al., 2002) as follows for instance with oxygen isotopes and $SO_2$ oxidation in sulfates:

$^{17/18}\beta_{SO4-SO2} = \ln{^{17/16}\alpha_{SO4-SO2}} / \ln{^{18/16}\alpha_{SO4-SO2}} \sim (1/m_{16} - 1/m_{17}) / (1/m_{16} - 1/m_{18}) \sim 0.530$

with $^{17/18}\beta_{SO4-SO2}$, the mass exponent describing the relative fractionation between $^{17}O/^{16}O$ and $^{18}O/^{16}O$, m, the mass of each isotope and $^{17/16}\alpha_{SO4-SO2}$, the isotope fractionation factor between two phases (defined in the introduction). Same equations can be written for $^{33}S$ and $^{36}S$.

The high temperature approximation has been shown to be applicable for a wide range of temperature phases

(Dauphas and Schauble, 2016) and isotope systems (S, Fe, Mg, O, Si…). Mass exponent value then depends on the specific reaction considered but usually, $^{17}\beta$, $^{33}\beta$ and $^{36}\beta$-values used are close to 0.524, 0.515 and 1.889 respectively. Thus, the β-exponent represents the slope in a δ-δ space, that is the mass-dependent fractionation line. More rigorously, the trend of isotopic variations is a curve, which is often approximated by a straight line. In this paper, we do not use this simplification. Deviation from the reference "mass-dependent" curve results

therefore in an anomalous isotopic composition, quantified by the Δ-parameter following Eq. (1), Eq. (2) and Eq. (3) (Farquhar and Wing, 2003; Thiemens, 1999):

$$\Delta^{17}O = \delta^{17}O - 1000 \times [(\delta^{18}O / 1000 + 1)^{0.524} - 1] \qquad (1)$$

$$\Delta^{33}S = \delta^{33}S - 1000 \times [(\delta^{34}S/1000 + 1)^{0.515} - 1] \qquad (2)$$

$$\Delta^{36}S = \delta^{36}S - 1000 \times [(\delta^{34}S/1000 + 1)^{1.889} - 1] \qquad (3)$$





Small non-zero $\Delta^{17}O$-$\Delta^{33}S$-$\Delta^{36}S$-values (typically between - 0.1 and + 0.1 ‰) can result from mixing, mass-dependent processes such as Rayleigh distillation or mass conservation effects and non-equilibrium processes (Farquhar et al., 2007; Ono et al., 2006) whereas large non-zero $\Delta^{17}O$-$\Delta^{33}S$-$\Delta^{36}S$-values (higher than + 0.2 ‰ or lower than - 0.2 ‰) imply mass-independent fractionation (Cabral et al., 2013; Delavault et al., 2016; Farquhar et al., 2000; Farquhar et al., 2007b; Farquhar et al., 2002; Ono et al., 2003). Oxidation reactions would then change

$\delta^{17}O$ and $\delta^{18}O$ but not the $\Delta^{17}O$, which would only vary through mixing of O-reservoirs with variable $\Delta^{17}O$. Possible mechanisms producing non-zero $\Delta^{17}O$-$\Delta^{33}S$-$\Delta^{36}S$-values in sulfate aerosols are discussed in the following sections. In this paper, we investigate the different processes responsible for the $\Delta^{17}O$, $\Delta^{33}S$ and $\Delta^{36}S$ recorded by black crusts sulfates and what can be inferred in terms of black crust formation.

## 3. Sampling and Methods

### 3.1. Sampling sites

To access sulfate aerosols from the Parisian region, black crusts were sampled following the prevailing winds according to a NW-SE cross-section, from Fécamp to Sens (Fig. 2 b, c). Therefore, the studied area covers rural,

urban and industrial zones including four power plants, major highways and the large Paris metropolis.

A total of 27 samples were collected on the external face of churches, monuments and on walls in the streets. The substrates were generally Lutetian and Cretaceous limestone, the typical building rocks in the Parisian Basin. To ensure a representative sample of sulfate aerosols, the sampling was carried out preferentially oriented to NW or, if possible, not directly exposed to traffic emission. Moreover, to avoid sulfate contamination from soils (i.e. salts

by capillary action, water from run-off…), black crusts were sampled at least at a height of 1.50 m above ground level. More details about samples are summarized in Table 1.

### 3.2 Methods

X-ray diffractometry (XRD D2-phaser BRUCKER, ISTeP Sorbonne Université) was used to specify the

mineralogical nature of each sample and therefore, to demonstrate the nature of sulfur. Structural and chemical aspects were subsequently investigated using Scanning Electron Microscopy (SEM, ISTeP Sorbonne Université). Sulfates were leached from 20-100 mg of black crusts and the conversion of gypsum into pure barite was performed according to the protocol developed at the Institut des Sciences de la Terre de Paris (ISTeP) as described by Le Gendre et al. (2017). The use of an ion-exchange resin in this protocol enables the concentration

and separation of sulfates from other compounds such as nitrates that can affect the O-isotopic measurements. From about 3 mg of the pure barite samples, the sulfate O-isotopic ratios were measured using the laser fluorination line coupled to a Delta V IR-MS at the Institut de Physique du Globe de Paris (IPGP) (Bao and Thiemens, 2000). Due to $SO_2F_2$ formation during $BaSO_4$ fluorination, that leads to incomplete $O_2$ extraction, measured $\delta^{18}O$ and $\delta^{17}O$ are fractionated but were corrected as deduced from the analysis of the international

barite standard NBS127 ($\delta^{18}O$ = 8.6 ‰, $\Delta^{17}O$ ~ 0 ‰); no correction was applied on $\Delta^{17}O$, remaining unchanged (Bao and Thiemens, 2000). For two NBS127 measured each day during five days (n=10), we obtained a mean $\delta^{18}O$ = -0.43 ± 0.54 (2σ) and a mean $\Delta^{17}O$ = 0.044 ± 0.020 (2σ) within error of the recent value reported by



Cowie and Johnston (2016). Thus, a correction factor of 9.03 was applied for $\delta^{18}O$ for all analyzed samples based on the certified value of NBS127.

The remaining $BaSO_4$ was reduced to hydrogen sulfide ($H_2S$) by reaction during 2 hours with a heated mixture of hydrochloric (HCl), hydroiodic (HI) and hypophosphorous ($H_3PO_2$) acids following the protocol described in Thode et al. (1961). $H_2S$ was purged and precipitates into silver sulfide ($Ag_2S$) passing through a silver nitrate ($AgNO_3$) solution. $Ag_2S$ was then converted to $SF_6$ and purified (Ono et al., 2006b) and quantified before being analyzed by isotope ratio mass spectrometry (Thermo-Fisher MAT-253) at the GEOTOP, Université du Québec à

Montreal (UQAM). The $\delta^{34}S$-values are expressed versus V-CDT assuming a $\delta^{34}S_{S1}$= -0.3 ‰ vs CDT isotope composition. Our data were then expressed against CDT, using previous analyses in our laboratory (Defouilloy et al., 2016; Labidi et al., 2012), following the method described by Defouilloy et al. (2016). Our analysis of the IAEA-S1 (n = 8) yielded: $\delta^{34}S$ = -0.29± 0.04 ‰, $\Delta^{33}S$ = 0.080± 0.010 ‰ and $\Delta^{36}S$ = -0.852 ± 0.085 ‰ vs V-CDT. Analysis of the IAEA-S2 (n = 8) gave: $\delta^{34}S$ = 22.49± 0.26 ‰, $\Delta^{33}S$ = 0.025± 0.005 ‰ and $\Delta^{36}S$ = -0.196±

0.223 ‰ vs V-CDT. Analysis of the IAEA-S3 (n = 8) gave: $\delta^{34}S$ = -32.44 ± 0.30 ‰, $\Delta^{33}S$ = 0.069 ± 0.023 ‰ and $\Delta^{36}S$ = -0.970 ± 0.277 ‰ vs V-CDT. All values are within the ranges of $\delta^{34}S$, $\Delta^{33}S$, $\Delta^{36}S$ accepted or measured by other laboratories for these international standards (Au Yang et al., 2016; Farquhar et al., 2007b; Labidi et al., 2014; Ono et al., 2006b). Analysis of the international sulfate standard NBS-127 was also performed and gave a $\delta^{34}S$ of 20.8 ± 0.4 ‰ (2$\sigma$; n=12), consistent with the 20.3 ± 0.4 ‰ value reported by the IAEA.


## 4. Results

### 4.1. Morphological and chemical aspects

After having confirmed the gypsum nature of the sample by X-ray diffraction, the structural and chemical aspects

of black crusts from four different environments were investigated on the basis of SEM observations. In agreement with previous studies (Fronteau et al., 2010; Siegesmund et al., 2007), all samples display two distinct layers. An opaque layer (few tens of $\mu m$) comprising massif and sparse gypsum crystals as well as aggregates of clay minerals and particulate matter overlies a layer (~100 $\mu m$) composed of more crystallized acicular and rosette-like crystals gypsum (tens of $\mu m$, Fig. 3a). As shown on Fig. 3b, soot is both present in urban and rural

encrustations consistent with previous observations (Guo et al., 2010). Moreover, fly ash particles resulting from coal or oil combustion are present in all environments. Parisian samples (PA13-2 and PA14-1) show many fly ashes of a diameter size < 10 $\mu m$ (primarily composed of Fe) with small gypsum crystals (few micrometers) on their surfaces (Fig. 3c). This is consistent with the catalyzer effect of combustion particles released by diesel and gasoline vehicles, which increases the rate of $SO_2$ fixation as sulfate (Rodriguez-Navarro and Sebastian, 1996).

Scarce fly ashes were also observed in samples from the city of Mantes-la-Jolie (northwest of Paris). The sample MR27-1 shows isolated halite crystals (< 10 $\mu m$, Fig. 3e) which can result from marine aerosols, in agreement with its location near the sea but occurrence of numerous fly ashes (Fig. 3d), most likely from power plants and traffic roads highlight that even the more rural environments are not free of anthropogenic emissions. The dissolution of rhomboedric calcite and subsequent precipitation of gypsum crystals can also be observed on Fig.

3f.





In summary, the presence of particulate matter and salts highlights several local or distant sources of S-bearing compounds and a prevailing anthropogenic source in the whole Parisian basin atmosphere, which can be distinguished and quantified with the isotopic composition of sulfate.


### 4.2. Isotopic composition of black crusts sulfates

The sulfur and oxygen isotopic compositions of black crusts sulfates are reported in Table 2. The $\delta^{18}O$ and $\delta^{34}S$ values cover a wide range from 7.5 to 16.7 ‰ ± 0.5 ‰ (2σ) and from -2.7 to 14.0 ‰ ± 0.2 ‰ (2σ) with a mean of 11.3 ± 2.4 ‰ and 3.8 ± 4.8 ‰ respectively. All samples have positive $\Delta^{17}O$ values, ranging from 0.08 to 2.56 ‰ ± 0.05 ‰ (2σ) with an average value of 0.86 ‰. Furthermore, it is noteworthy that 67 % of black crusts samples have $\Delta^{17}O$ > 0.65 ‰. The $\Delta^{33}S$ and $\Delta^{36}S$ are both negative and vary between -0.34 and 0.00 ± 0.01 ‰ and between -0.7 and -0.2 ± 0.2 ‰ (2σ) respectively. No obvious correlation exists between neither $\delta^{18}O$, $\Delta^{17}O$, $\delta^{34}S$, $\Delta^{33}S$ and the distance from coastline (Fig. S1).

## 5. Discussion

### 5.1. The $\delta^{34}S$-$\delta^{18}O$ systematic as a sulfate sources tracer

Sulfate in black crusts may have multiple origins that could be either primary and/or secondary. We refer to primary sulfates here, as sulfates that are not formed in the atmosphere from $SO_2$-oxidation. These ones can originate from the host-rock itself where sulfur occurs both as sulfide such as pyrite that is subsequently dissolved and oxidized as sulfate, and as carbonate-associated sulfates (CAS), which substitute for carbonate in the lattice. Sulfates are also directly emitted into the atmosphere for instance by sea-spray, resulting in sea-salt sulfate aerosols, or as products of combustion by refineries, vehicle exhaust or biomass burning; these commonly correspond to "primary sulfates" in the literature. On the contrary secondary sulfates result from the oxidation of tropospheric S-bearing gases (mainly $SO_2$) and other compounds including Dimethyl sulfide (DMS, $(CH_3)_2S$) by various oxidants ($O_3$, $H_2O_2$, OH, $O_2$-TMI, $NO_2$, etc…). As black crusts are mainly constituted of gypsum ($CaSO_4$, $2H_2O$), the $\delta^{34}S$ associated with the $\delta^{18}O$ are often used to trace the sources of the initial $SO_2$ from which the sulfate and then black crusts formed. Previous works discussed the distinction between natural and anthropogenic sources (Montana et al., 2012; Montana et al., 2008) as well as between extrinsic (atmospheric) and intrinsic (mortars, plasters) sulfates (Vallet et al., 2006; Klemm & Siedel, 2002; Kloppmann et al., 2011). Sulfates from black crusts analyzed in this study have oxygen and sulfur isotopic compositions that overlap other black crusts from Europe (Longinelli & Bartelloni, 1978; Torfs et al., 1997, Kramar et al., 2011, Vallet et al., 2006). In particular, there is a positive correlation between $\delta^{34}S$ and $\delta^{18}O$ covering a large range of variation of ~17 and ~9 ‰ respectively, which can be interpreted in two ways: either a process leads to a variable enrichment or depletion of $^{18}O$ and $^{34}S$ in the crusts and/or it reflects a mixing between at least one depleted (in both $^{18}O$ and $^{34}S$) and one enriched end-member (Fig. 4). With a fractionation factor for $^{18}O/^{16}O$ between the dissolved sulfate and the gypsum, $CaSO_4$, ~ 1.002 or 1.0036 (experimental and natural values respectively) (Lloyd, 1968) and a fractionation factor for $^{34}S/^{32}S$ ranges between 1.000 and 1.0024 (Ault and Kulp, 1959; Nielsen, 1974; Raab and


Spiro, 1991; Thode et al., 1961), black crust precipitation would fractionate both O and S isotopes of sulfate. Therefore, if the gypsum precipitation follows a Rayleigh-type process where residual dissolved sulfates are leached, the $\delta^{34}S$-$\delta^{18}O$ co-variation should follow a slope of 0.67 ± 0.02 (as deduced from the range of fractionation coefficients given above; Fig. 4). However, the slope defined by the samples is steeper, around 1.52 ($R^2$ = 0.58) implying that the gypsum precipitation is not the main mechanism driving $\delta^{34}S$ and $\delta^{18}O$ variations.

Another process that could affect $\delta^{34}S$-$\delta^{18}O$-values is the partial oxidation of $SO_2$ by different atmospheric oxidants (e.g. $O_2$-TMI, $H_2O_2$, $O_3$, OH). Using the fractionation factors $^{34}\alpha_{SO4-SO2}$ obtained experimentally at 19°C by Harris et al. (2012a) for each oxidant ($^{34}\alpha_{SO4-SO2}$ (OH) = 1.0113 ± 0.0024; $^{34}\alpha_{SO4-SO2}$ ($H_2O_2$) = 1.0151 ± 0.0013; $^{34}\alpha_{SO4-SO2}$ ($O_2$-TMI) = 0.9894 ± 0.0043; $^{34}\alpha_{SO4-SO2}$ ($O_3$) = 1.0174 ± 0.0028) and their respective proportions modeled by Sofen et al. (2011), usually cited in the literature for present day atmosphere (27 % OH; 18 % $O_2$-

TMI; 50 % $H_2O_2$; 5 % $O_3$), we calculated a global fractionation factor of 1.0097. Consequently, following a Rayleigh distillation model and an initial $SO_2$ with $\delta^{34}S$ = 0 ‰, the cumulated sulfates would be enriched ~ 9 ‰ at maximum when < 10 % $SO_2$ is oxidized, which cannot explain the ~ 17 ‰ variation in $\delta^{34}S$, especially since 40 % oxidized $SO_2$ is reported (Chin et al., 2000). To generate a fractionation as high as 17 ‰, $O_3$ and $H_2O_2$ oxidation pathways should increase drastically (i.e. with the absence of $O_2$-TMI pathway), a scenario not realistic,

implying also an increase of $\Delta^{17}O$-values up to ~ 6.5 ‰ with $\delta^{34}S$, which is not consistent with our data ($\Delta^{17}O$ ~ 0 ‰, see section 5.2, Fig. 5). Therefore, $SO_2$ partial oxidation could explain a part of the data but not the whole isotope variations. The large $\delta^{34}S$ range could also reflects temporal variation, since $\delta^{34}S$ in Greenland ice cores was higher than 10 ‰ before the Industrial period (Patris et al., 2002), mainly dominated by $SO_2$ from DMS (Sofen et al., 2011) and then decreased below 4 ‰ in the 1960's, dominated by anthropogenic $SO_2$. Following

this variation, black crusts on churches recently renovated should display low $\delta^{34}S$ and churches renovated before the Industrial period should display higher $\delta^{34}S$. However, Church Saint Aspais of Melun (ME77-2, $\delta^{34}S$ = -0.538 ‰) and Cathedral Notre-dame of Évreux (EV27-1, $\delta^{34}S$ = 6.597 ‰) restored after World War II compared to Church Notre-Dame of Pont sur Yonne restored for the last time in 1772 (PY89-1, $\delta^{34}S$ = 0.462 ‰) present no significant temporal variation, that might be due to higher proportions of anthropogenic $SO_2$ emitted since

recently (0.5 Tg S.yr$^{-1}$ before Industrial period and 67 Tg S.yr$^{-1}$ at present day; Sofen et al. (2011) and references therein). Thus, black crusts do not seem to record temporal variation, even if samples with $\delta^{34}S$ = -2.663 and $\delta^{34}S$ = 13.99 ‰ should be dated to confirm this assumption. Alternatively, with well-exposed surfaces to precipitation emphasizing wash-out and subsequent reprecipitation, black crusts could rather probe the "recent" $SO_2$-oxidation. If $\delta^{34}S$-$\delta^{18}O$ reflect mixing, at least two end-members are required. A first one would be enriched with $\delta^{34}S$ and

$\delta^{18}O$ both around 18 ‰, which in view of the sampling cross-section from NW to SE and west-dominating winds could correspond to the sea-sprays isotopic signature. This is however not consistent with available data showing that sea-sprays usually displays a $\delta^{18}O$ ~ 9 ‰ (Markovic et al., 2016) and a $\delta^{34}S$ ~ 21 ‰ (Rees et al., 1978). Furthermore, the amount of emitted DMS produced by phytoplankton and oxidized in the atmosphere (11-25 TgS.yr$^{-1}$) being higher than sea-salt emissions (6-12 TgS.yr$^{-1}$ (Alexander et al., 2005) and references therein)),

with a high $\delta^{34}S$ of 15-20 ‰ (Calhoun et al., 1991), sulfate aerosols deriving from DMS oxidation could rather represent this $^{18}O$-$^{34}S$-enriched end-member. However, the absence of correlation with the distance from coastline (Fig. S1c) and $\Delta^{17}O$-values near zero for high $\delta^{34}S$-values (Fig. 5) are not consistent with significant DMS





contribution, that are mostly oxidized by ozone having $\Delta^{17}O$ = 8.75 ‰ (Alexander et al., 2012) (see section 5.2).

Despite some isolated halite crystals were observed in one sample (Fig. 3e), we conclude that, overall, marine

aerosols (DMS and sea-salt sulfates) do not relate to the high $\delta^{34}S$-$\delta^{18}O$-end-member. Major element contents

(e.g. Na, Cl) have not been measured here even if they could further constrain and quantify the presence of

marine aerosols. The structural analyses of black crusts emphasize dissolution of the underlying carbonate.

Consequently, carbonate-associated sulfates (CAS; with S abundances varying between a few tens to thousand

ppm (Kampschulte and Strauss (2004) and references therein)) of the building stone would also be dissolved and

reprecipitated in black crusts and may well represent the enriched end-member. CAS analyses from 3 different

sites in Atlantic and Pacific oceans over the last 25 Myr and in the Umbria-Marche Apennines formation in

central Italy recording middle Cretaceous Tethys ocean signatures show $\delta^{34}S$-values from 11 to 24 ‰ with $\delta^{18}O$

from 5 to 21 ‰ similar to barite isotopic composition (Rennie and Turchyn, 2014; Turchyn et al., 2009). CAS

would thus perfectly match this end-member. Plaster used to seal blocks of carbonate stones are made through

Lutetian gypsum dehydration and could also well represent the $^{18}O$-$^{34}S$-rich end-member. Kloppmann et al.

(2011) measured the O and S isotopic compositions of black crusts, mortars and plasters from several French

churches and castle. With $\delta^{34}S$-values varying between 12.6 and 18.3 ‰ and $\delta^{18}O$-values around 14.6 and 21.5

‰, it appears that the plaster would also represent a matching end-member. Coming from the host rock or plaster,

this enriched end-member (CAS/PL on Fig. 4) has a $\Delta^{17}O \sim 0$ ‰, consistent with the absence of atmospheric

sulfate aerosols ($\Delta^{17}O > 0$ ‰), a conclusion in agreement with that of Kloppmann et al. (2011).

The depleted end-member would be characterized by $\delta^{34}S$ below -3 ‰ with little constrained $\delta^{18}O$, from 5 to 15

‰. Sedimentary pyrites contained in the building carbonate stone are known to have $\delta^{34}S < 12$ ‰ (since at least

the last 500 Myr, (Canfield, 2004). Their dissolution and oxidation by rainwater produce sulfates that can be

incorporated into black crusts. Despite a sulfide content that can vary between a few tens to a thousand ppm

(Thomazo et al., 2018), our sampled carbonate stones are very whitish, suggesting a low sulfide content. Thus,

even if it would certainly not affect the mass balance, we took pyrite oxidation into account, as did other studies

on black crusts (Kramar et al., 2011; Vallet et al., 2006). Besides, the sulfur isotope fractionation factor during

pyrite oxidation being negligible (between 0.996 and 1; Thurston et al. (2010) and references therein) compared

to oxygen isotopes, we modeled the $\delta^{18}O$ variation according to a Rayleigh distillation to represent the sulfide

oxidation at the atmosphere-carbonate building stone interface. With an initial $\delta^{18}O \sim -6$ ‰ of rainwater in Paris

basin and a mean fractionation factor between water and sulfates of 1.010 (Gomes and Johnston, 2017), sulfates

from pyrite oxidation would have $\delta^{18}O \sim 4$ ‰ and as low as -6 ‰ if water would be in limited amounts (i.e.

residual fraction of water F $\sim$ 0). Thus, the depleted end-member could result from sedimentary pyrite oxidation.

However, our data are strikingly higher than for black crusts from Ljubljana (Slovenia; (Kramar et al., 2011)),

which show $\delta^{34}S$ as low as -20 ‰ and $\delta^{18}O$ between -2 and 5 ‰ (Fig. 4) that would be typical for pyrite

oxidation. This model would also not be able to account for non-zero $\Delta^{17}O$ discussed in the following section.

This means that another source should have negative $\delta^{34}S$. Anthropogenic sulfur represent $\sim$ 60 % of the total

sulfur emitted worldwide and includes combustion products from oil, coal as well as biomass and thus display

both large $\delta^{34}S$ and $\delta^{18}O$ variations. The combustion of herbs and diesel results in $\delta^{34}S$ and $\delta^{18}O$ values varying



between 9.55 and 16.42 ‰, and between 5.5 and 10.5 ‰ respectively (Lee et al., 2002). A lower range of $\delta^{34}S$ was measured for sulfur emitted by transport, incinerator and cement factories in Paris (between -0.57 and 11.33 ‰; unpublished data) and a narrow range between -2.1 and 2.8 ‰ was reported for vehicles emissions (Torfs et al., 1997). However, oxygen isotopic compositions were not measured in these two studies. When considering

coal and oil combustion, $\delta^{34}S$-values can be as low as -10 ‰ (Nielsen, 1974) while Faure (1986) reported a wide range between -30 and 30 ‰, and between -8 and 32 ‰ respectively. Therefore, the depleted end-member with a negative $\delta^{34}S$-value below -3 ‰ could be typified by anthropogenic sulfur emissions (An on Fig. 4). This is in agreement with the variable non-zero $\Delta^{17}O$-values (see section 5.2). Because sulfate aerosols can be either primary or secondary with various $SO_2$ oxidation pathways having distinct $\delta^{18}O$-$\Delta^{17}O$-values and O-fractionation

factors, atmospheric aerosols would result in variable $\delta^{18}O$-values. So far, we assumed only two end-members even if the scatter trends may actually allow more mixing end-members. Consequently, the observed $\delta^{34}S$-$\delta^{18}O$ correlation could reflect a mixing between a little variable high $\delta^{34}S$ and $\delta^{18}O$ end-member with a low $\delta^{34}S$ and variable low $\delta^{18}O$ end-member reflecting the different $SO_2$ oxidation channels in the atmosphere. In view of the $\delta^{18}O$ variability, mixing proportions of CAS/PL and An end-members were calculated based only on $\delta^{34}S$-values.

We chose a CAS/PL $\delta^{34}S$-value of 18 ‰, in the range from 11 to 24 ‰ (Kloppmann et al., 2011; Rennie and Turchyn, 2014; Turchyn et al., 2009)(Fig. 4) and an An $\delta^{34}S$-value of -3 ‰, similar to Montana et al. (2012). Using a mass balance calculation, CAS/plaster proportions range between 2 and 81 % with an average ~ 30 %. Therefore, the host rock sulfate is on average not the main S-provider, highlighting atmospheric sulfate aerosols sampling by black crusts. In order to go further in the anthropogenic sources (primary and/or secondary sulfate

aerosols) and oxidation pathways characterization, we combined, in the following section, the measured $\Delta^{17}O$ to $\delta^{34}S$-$\delta^{18}O$ systematic.

### 5.2. The $\Delta^{17}O$ values as a proxy for $SO_2$ oxidation pathways

Figure 5 shows samples having near zero $\Delta^{17}O$-values with $\delta^{34}S$-values from -3 up to 14 ‰ and $\Delta^{17}O$-values up to

2.6 ‰, with $\delta^{34}S$-values that are < 10 ‰ when $\Delta^{17}O$ > 1 ‰. These values are consistent with $\Delta^{17}O$-values of sulfate aerosols worldwide, that range from 0.14 to 3 ‰ with a mean ~ 0.78 ‰, collected in rainwater or on filters in La Jolla (USA), Baton Rouge (USA), Bakerfield (USA), White Mountain Research Station (WMRS, USA), Wuhan (China) (Bao et al., 2001a; Jenkins and Bao, 2006; Lee and Thiemens, 2001; Li et al., 2013; Romero and Thiemens, 2003). The lack of correlation between $\Delta^{17}O$ and the distance from coastline (Fig. S1b) is

consistent with the evidence that $\delta^{34}S$-values are also not correlated with distance from coastline, confirming that DMS, which is mostly oxidized by ozone and carries high $\Delta^{17}O$-values (Alexander et al., 2012), is less significant compared to anthropogenic sulfur.

Large positive $\Delta^{17}O$ anomalies in sulfate aerosols are inherited from their atmospheric oxidants, that were ultimately produced during $O_3$-photochemically induced genesis. In theory, other mechanisms exist such as

magnetic isotope effect (see section 5.3.2) but have not been recognized yet. Sulfur dioxide will undergo oxidation via OH radical (in gaseous phase) or $H_2O_2$, $O_3$ or $O_2$ catalyzed by TMI (in aqueous phase) as well as other potential oxidants such as $NO_2$ or Criegee radicals, in the troposphere. As $NO_2$ and Criegee radicals are minor species, these have accordingly been less studied with respect to $\Delta^{17}O$ and are generally omitted. Resulting





from photochemical reactions, $O_3$ molecules possess oxygen-MIF compositions with $\Delta^{17}O \sim 35$ ‰ (Janssen et al.,
1999; Lyons, 2001; Mauersberger et al., 1999) and lower in the troposphere ~26 ‰ (Vicars and Savarino, 2014).
Every molecule inheriting oxygen atoms from ozone will also have positive $\Delta^{17}O$ including $H_2O_2$ with an average
$\Delta^{17}O \sim 1.3$ ‰ (Savarino and Thiemens, 1999). OH, which isotopically exchanges with water vapor, and $O_2$-TMI
have mass-dependent composition with $\Delta^{17}O \sim 0$ (Dubey et al., 1997; Holt et al., 1981; Lyons, 2001) and ~ -0.34
‰ (Barkan and Luz, 2005) respectively. Savarino (2000) measured the O-isotopic compositions of sulfates
derived from these different oxidation pathways and showed that the OH and $O_2$-TMI oxidation channels do not
result in a mass-independent fractionation ($\Delta^{17}O = 0$ and -0.09 ‰ respectively) whereas $O_3$ and $H_2O_2$ radicals
transfer ¼ and ½ respectively of their isotopic anomaly to the sulfate thus resulting in a mass-independent
fractionation ($\Delta^{17}O = 8.75$ ‰ and 0.65 ‰ respectively) (e.g. Bao et al., 2001a; Bao et al., 2000; Bao et al., 2001b;
Bao et al., 2010; Jenkins and Bao, 2006; Lee et al., 2002; Lee and Thiemens, 2001; Li et al., 2013; Martin et al.,
2014). Mass-dependent isotopic fractionation during $SO_2$ oxidation may change $\delta^{17}O$ and $\delta^{18}O$ but not the $\Delta^{17}O$
that only depends on the mixing of O-reservoirs with variable $\Delta^{17}O$. Therefore, samples with low $\delta^{34}S$-values,
consisting mostly of anthropogenic sulfur (section 5.1), and $\Delta^{17}O > 0.65$ ‰ obviously point to a significant
anthropogenic $SO_2$ fraction oxidized by $O_3 + H_2O_2$ or by $O_3$ and to a lesser extent by $O_2$ TMI and OH, depending
on the water pH (Lee and Thiemens, 2001) and correspond to samples with significant atmospheric sulfate
aerosols. As the combustion does not produce sulfates with mass-independent signatures (Lee et al., 2002),
primary sulfate aerosols have $\Delta^{17}O \sim 0$ ‰, sea-salt sulfates as well, thus samples with $\Delta^{17}O < 0.65$ ‰ and low
$\delta^{34}S$-values then either represent primary anthropogenic sulfate aerosols and/or $SO_2$ oxidized by OH or $O_2$-TMI
and/or a subtle mixing of oxidants to yield near-zero $\Delta^{17}O$ (Fig. 5). Samples having the highest $\delta^{34}S$ values were
identified as being representative of host-rock sulfates (CAS/PL end-member) and their $\Delta^{17}O$-values near zero is
in agreement with this origin. Indeed, marine sulfates can have $\Delta^{17}O$-values down to -0.70 ‰ in the geological
record (a consequence of high $pCO_2$) during the Marinoan, ~ 635 Myr ago, but most of the time, $\Delta^{17}O$-values are
typically around 0 and > -0.2 ‰, implying a lower $pCO_2$ and/or a higher flux of sulfide re-oxidation in sediments
(Bao et al., 2008).

Our $\Delta^{17}O$-values representing the first data in black crusts, it could be speculated that some unexpected processes
such as magnetic isotope effects (see section 5.3.2) occurred during black crust formation, i.e. as opposed to
$\Delta^{17}O$-anomaly being inherited from $SO_2$ oxidants. However, it is worth noting that Lee et al. (2002) also
measured the O-multi isotopic compositions of sulfate aerosols (i.e. from the atmosphere as opposed to reaction
on a solid substrate) from Paris and obtained $\Delta^{17}O = 0.2$ and 0.8 ‰ for the Paris highway and in the 13[th] zone
respectively, which is in good agreement with our three samples collected in Paris (from 0.17 to 0.89 ‰). Thus,
this is consistent with black crusts formation recording mostly an atmospheric signal and no significant magnetic
isotope effect on $\Delta^{17}O$.

The minimum proportion of secondary sulfate aerosols in black crusts can be quantified using the $\Delta^{17}O$-values of
each oxidant. First, we excluded samples that are the most contaminated by the host rock as inferred from the
mixing between CAS/plaster and An based on the $\delta^{34}S$-values (see previous section) and are therefore less
susceptible to record sulfate aerosols. As sample YV76-1 contains ~ 65 % of CAS/plaster but has still a mass-
independent $\Delta^{17}O$-value of 1 ‰, it defines the lower bound proportion where anomalous $\Delta^{17}O$ is still preserved.





Thus, samples BR91-1, TV27-1 and TO77-1 having more than 65 % of CAS/plaster were not taken into account. Assuming that samples having $\Delta^{17}O > 0.65$ ‰ obviously represent secondary sulfate aerosols resulting from $SO_2$

oxidation by $O_3$ and $H_2O_2$, the minimum proportion of MIF-bearing sulfates is ~ 63 % which is close to ~ 54 % for La Jolla rainwater (calculated with a higher $\Delta^{17}O$ of 0.85 ‰ in Lee and Thiemens (2001) and recalculated here for $f_{MI} + f_{MD} = 1$ where MI and MD denote mass-independent and mass-dependent fractionation respectively) and ~ 55 % modeled by Sofen et al. (2011). Considering now $\Delta^{17}O_{O3} = 8.75$ ‰ and $\Delta^{17}O_{H2O2} = 0.65$ ‰ of sulfates derived from the $SO_2$ oxidation by $O_3$ and $H_2O_2$ respectively, we can calculate the proportions (or

fluxes) of the two oxidation channels around 4 ($O_3$) and 96 % ($H_2O_2$), following mass balance Eq. (4) Lee and Thiemens (2001) :

$$\Delta^{17}O_{measured} = f(O_3) \times \Delta^{17}O_{O3} + f(H_2O_2) \times \Delta^{17}O_{H2O2} \qquad (4)$$

It is noteworthy that the estimated proportion of secondary sulfate in the black crust is highly dependent on the oxidation channel fluxes. For instance, a 5% increase in the $O_3$ oxidation channel flux decreases the secondary

sulfate proportion in the black crust by 30%. Therefore, even if precise quantification is not possible, we can assume that secondary sulfate aerosols, mostly formed by oxidation of $SO_2$ by $H_2O_2$, dominate in black crusts and hence result from aqueous phase reaction.

### 5.3. Black crusts S-MIF signature

#### 5.3.1 Processes implicated in black crust formation

The $\Delta^{17}O$-parameter provides key evidence for $SO_2$ oxidation by "atmospheric" oxidants but does not allow distinction between (1) $SO_2$ oxidized in the atmosphere generating secondary sulfate aerosols or (2) $SO_2$ deposited and oxidized on the building stone. This question can be addressed using $\Delta^{33}S$-$\Delta^{36}S$ systematics.

$\Delta^{33}S$-$\Delta^{36}S$-values recorded by black crust sulfates range between -0.34 and 0.00 ± 0.01 ‰ for $\Delta^{33}S$ and between -

0.7 and -0.2 ± 0.2 ‰ (2σ) for $\Delta^{36}S$ (Table 2). These values are quite unusual compared with anthropogenic and natural aerosols. As illustrated by Fig. 6, black crust sulfates $\Delta^{33}S$-values are all negative and it is worth noting that this depletion occurs with near constant $\Delta^{36}S$-values. This is somewhat distinct from most aerosols, which display almost exclusively positive $\Delta^{33}S$ up to ~0.5 ‰ and both positive and negative $\Delta^{36}S$ (Au Yang et al., 2019; Guo et al., 2010; Lin et al., 2018b; Romero and Thiemens, 2003; Shaheen et al., 2014). So far the only negative

$\Delta^{33}S$-values down to -0.6 ‰ were measured in sulfate aerosols from Beijing (China) during one winter month (Han et al., 2017), (no $\Delta^{36}S$-values provided) and these values were assumed to result from incomplete combustion of coal. This assumption ultimately relies on the work of Lee et al. (2002), which showed that primary anthropogenic aerosols formed by high temperature combustion (e.g. diesel) result in near-zero $\Delta^{33}S$-$\Delta^{36}S$-values whereas those formed by low temperature combustion (e.g. biomass burning) result in $\Delta^{33}S$ down to -

0.2 ‰ and $\Delta^{36}S$-values varying between -1.9 and 0.2 ‰ (data recalculated with $^{36}\beta = 1.9$). Negative $\Delta^{36}S$-values well correlated with biomass burning proxies are also reported in East China (Lin et al., 2018b) yet showed $\Delta^{33}S$ ~ 0 ‰. As many other cities, Paris has long been affected by coal and wood burning, we can hypothesize that $\Delta^{33}S$-$\Delta^{36}S$ variations result from high and/or low temperature combustion processes. Some black crust sulfates with near-zero $\Delta^{33}S$-$\Delta^{36}S$-values could result from high temperature combustion but this would not explain

negative $\Delta^{33}S$-$\Delta^{36}S$. Furthermore, according to Au Yang et al. (2016); Lin et al. (2018a); Lin et al. (2018b), low temperature combustion would preferentially fractionate $^{36}S$ over $^{33}S$, which should result in a steep slope in a





$\Delta^{33}$S-$\Delta^{36}$S space. The trend defined by our black crust samples shows higher $^{33}$S fractionation than $^{36}$S with $\Delta^{33}$S-values lower than that obtained by available low temperature combustion experiments (< -0.2 ‰) and with $\Delta^{36}$S-values in the range of aerosols. Furthermore, no $\Delta^{33}$S evolution is observed in black crusts sampled on churches

with different ages of renovation (see section 5.1; ME77-2 $\Delta^{33}$S = -0.210 EV27-1 $\Delta^{33}$S = -0.054 and PY89-1 $\Delta^{33}$S = -0.206) whereas we would expect a $\Delta^{33}$S-increase in black crusts from -0.2 ‰ and 0 ‰ due to the reduction of sulfur emission from low temperature replaced by high temperature combustion processes (Lee et al., 2002). Therefore, available data highlight that neither high nor low temperature combustion processes are responsible for low $\Delta^{33}$S measured in black crusts.

Part of black crust sulfates being atmospheric in origin, isotopic effects during $SO_2$ oxidation could be responsible for $\Delta^{33}$S-$\Delta^{36}$S variations. To better address this issue, we calculated the $\Delta^{33}$S-$\Delta^{36}$S-values of sulfates predicted by each of the main $SO_2$ oxidation pathways and by a mixing of them in the proportions given by Sofen et al. (2011). We used $^{33}\beta$ and $^{36}\beta$ determined by experiments of $SO_2$ oxidation by $O_2$-TMI, $H_2O_2$, $O_3$, OH (Harris et al. (2013b) and values cited in Au Yang et al. (2018); see caption text) and $NO_2$ (Au Yang et al., 2018) and T-

dependent equations determined by Harris et al. (2013b) to calculate each $^{34}\alpha$ with initial sulfur dioxide $\Delta^{33}$S and $\Delta^{36}$S of 0 ‰ (Lin et al., 2018b). As mentioned earlier (Au Yang et al., 2018; Harris et al., 2013b), none of these models can account for anomalous $\Delta^{33}$S-$\Delta^{36}$S values in either aerosols or in black crusts (Fig. 6). Their potential combination is also at odds with sulfate aerosols data from the literature (Au Yang et al., 2019; Guo et al., 2010; Lin et al., 2018b; Romero and Thiemens, 2003; Shaheen et al., 2014), resulting in negative $\Delta^{33}$S-$\Delta^{36}$S but not low

enough to explain $\Delta^{33}$S < -0.2 ‰. Available literature data are therefore not consistent with the anomalous $\Delta^{33}$S-$\Delta^{36}$S-values recorded in black crust sulfates.

Mass-dependent processes can also result in small $\Delta^{33}$S-$\Delta^{36}$S variations, depending on the magnitude of the $^{34}$S fractionation (Ono et al., 2006a). As briefly mentioned in section 5.2, a mixing between a $^{33,34}$S depleted end-member (An) consisting of anthropogenic sulfur ($\delta^{34}$S = -3 ‰, $\Delta^{33}$S = 0 ‰) and a $^{33,34}$S enriched sulfates end-

member (CAS/PL) from plaster or CAS ($\delta^{34}$S = 18 ‰, $\Delta^{33}$S = 0 ‰) (section 5.1.) would result in small $\Delta^{33}$S of -0.014 ‰ for 50 % mixing, which is far from the maximum measured $\Delta^{33}$S ~ -0.336. Moreover, the slope between $\Delta^{33}$S-$\Delta^{36}$S would be -7 at odd with our observations. Therefore, we conclude that mixing cannot account for the black crusts $\Delta^{33}$S-$\Delta^{36}$S variations.

**5.3.2 A new oxidation pathway implying magnetic isotope effect**
Several studies proposed that positive $\Delta^{33}$S measured in sulfate aerosols, with $\Delta^{33}$S up to +0.5 ‰, from e.g. East China and California result from stratospheric fallout of $SO_2$ (with $\Delta^{33}$S potentially up to +10 ‰ higher; Ono et al. (2013)), which underwent UV photolysis by short wavelength (Au Yang et al., 2016; Lin et al., 2018a; Lin et al., 2018b; Romero and Thiemens, 2003). This suggestion primarily relies on the similarities between $\Delta^{33}$S-$\Delta^{36}$S

values of sulfate aerosols and laboratory experiments of $SO_2$ photolysis conducted at different wavelengths (Romero and Thiemens, 2003) and on the correlation between $^{35}$S specific activity and $\Delta^{33}$S-values (Lin et al., 2018b). However, these studies never addressed the absence of the complementary negative $\Delta^{33}$S-reservoir, which is required to balance the positive $\Delta^{33}$S-reservoir. In this respect, it is worth mentioning that stratospheric aerosols trapped in Antarctic ice cores (see Gautier et al. (2018) and references therein) show both positive $\Delta^{33}$S

(up to ~ +2 ‰) and complementary negative $\Delta^{33}$S-values (down to -1 ‰) and weighed average $\Delta^{33}$S $\neq$ 0 ‰





explained by prior partial deposition. Stratospheric fluxes are actually too low to account for $\Delta^{33}S > 0.1$ ‰ (Au Yang et al., 2018). Accordingly some other authors rather tried to explain the positive anomalies of most aerosols with 'tropospheric' chemical reactions, that are $SO_2$ oxidation by the main oxidant including $NO_2$, $H_2O_2$, OH, $O_3$ and $O_2$-TMI, but experimental data results in $\Delta^{33}S$ were ~ 0 ‰ for all studied reactions (Au Yang et al., 2018;

Harris et al., 2013b). Isotope effects associated with $SO_2$-oxidation by minor species, such as Criegee radicals remains to be investigated (see Au Yang et al. (2018)) and it is also worth mentioning that previous experimental work mentioned above did not involve any 'tropospheric-type' photochemical reactions that could well be associated with mass-independent isotope effects. In summary, whatever the stratospheric vs. tropospheric origin of positive $\Delta^{33}S$-values recorded by most aerosols, there is a $^{33}S$-isotope imbalance and a missing reservoir with

negative $\Delta^{33}S$ that must exist. Han et al. (2017) reported $\Delta^{33}S$-values down to -0.6 ‰ in sulfate aerosols from Beijing. As discussed above, the author's suggestion calling for low temperature combustion is little supported by available data, and clearly the very restricted location and time interval - over a month - where these anomalies occurred cannot counter balance, both spatially and temporally, the common positive $\Delta^{33}S$-values of most aerosols; the missing reaction/reservoir requires, instead, to be ubiquitous worldwide.

In this study, black crust sulfates display negative $\Delta^{33}S$-values (from 0 ‰ down to -0.3 ‰). These values are certainly produced by chemical reactions, they would otherwise - according to the stratospheric origin - be the same as those measured among aerosols. Furthermore, the(se) chemical reaction(s) involved in the formation of black crusts must be distinct that those leading to the formation of tropospheric aerosols. As developed thoroughly, black crust could well represent the missing sulfur reservoir.

An additional observation is that negative $\Delta^{33}S$-values occur with near constant $\Delta^{36}S$ (from -0.8 to -0.2 ± 0.2 ‰; Fig. 6). This signature is typical of magnetic isotope effects (MIE), that involve a radical pair, where coupling between the nuclear magnetic moment of the nucleus of odd isotopes and the electron occurs, allowing for electron spin transition from singlet to triplet (or vice-versa) (Buchachenko et al., 1976). This leads to distinct half-lives between odd and even isotopes resulting in specific odd over even isotope enrichment (or depletion).

MIE has been so far reported for various reactions (Buchachenko, 2001, 2000; Turro, 1983) like sulfate thermochemical reduction (Oduro et al., 2011) or Fe reduction in magneto-tactic bacteria (Amor et al., 2016) for the most geologically relevant.

Review of available literature shows that either product or residue could be enriched in odd isotope depending

whether the radical pair is in triplet or singlet state (converting to singlet and triplet respectively). For more complex reactions, such as the UV photolysis of phenacylphenylsulfone (Kopf and Ono, 2012) the product can be enriched at the beginning of the reaction and become depleted as the reaction proceeds; the authors hypothesized a series of combined reactions involving at least two radical pairs produced through a series of distinct chemical pathways. Even if this latter reaction requires photochemistry, MIE is not necessarily produced by photochemical

reaction (Amor et al., 2016; Oduro et al., 2011).

Magnetic effect could occur in the atmosphere during aerosols formation (being photochemically-induced or not), leading to residual $^{33}S$-depleted atmospheric $SO_2$ from which black crusts would subsequently formed. This model would however predict some sulfate aerosols formed subsequently to display negative $\Delta^{33}S$: such values are extremely uncommon being primarily restricted to the Beijing winter month (Han et al., 2017). Instead,



magnetic effect could occur during black crust formation (being photochemically-induced or not), leading to residual $^{33}$S-enriched atmospheric $SO_2$ from which tropospheric aerosols would subsequently formed; which is consistent with available observations. This model would however predict some black crust formed subsequently to display positive $\Delta^{33}$S: such values have not been found yet and this may well reflect sample bias, our data being the first reported for such samples. Both scenario imply non-zero $\Delta^{33}$S of residual atmospheric $SO_2$ which

contrast with the data by Lin et al. (2018b) showing $\Delta^{33}$S ~ 0 ‰ (n = 5, $\Delta^{33}$S varying from -0.04 to 0.01 ± 0.01 ‰). Given that, in the study of Lin et al. (2018b), $SO_2$ was sampled close to the third largest Chinese megacity, such non-zero $\Delta^{33}$S-values may thus be rather symptomatic of emitted (i.e. anthropogenic) $SO_2$ rather than residual/background (i.e. after significant black crust and aerosols formation); this is consistent with the observation that non-zero $\Delta^{33}$S-values of residual/background atmospheric $SO_2$ are erased by anthropogenic $SO_2$

having zero $\Delta^{33}$S-values (Au Yang et al., 2019) moving towards the local source(s) of anthropogenic $SO_2$.

In the absence of additional observations, proposing a chemical reaction would be very speculative, but our data clearly point towards the occurrence of magnetic effect occurring during the formation of black crust, involving ubiquitous heterogeneous chemical reactions. This is supported by previous recognition of sulfur radicals such as $SO_x^-$ (Herrmann, 2003). Clearly, the reaction does not occur after sulfate formation such as during

dissolution/precipitation mechanisms, which does not involve any radical species. As mentioned above, magneto-tactic bacteria can produce MIE when reducing Fe (Amor et al., 2016). Microbial activity being sometimes present on black crusts (Gaylarde et al., 2007; Sáiz-Jiménez, 1995; Scheerer et al., 2009; Schiavon, 2002; Tiano, 2002), we cannot rule out a MIE caused by micro-organisms. Another implication that can be tested in future work is that the kinetics of heterogeneous reactions leading to sulfate and black crust formation should be

comparable or faster than those leading to aerosol formation. So far, Li et al. (2006) showed comparable loss of atmospheric $SO_2$ by heterogeneous oxidation on calcium carbonate substrate and by gas phase oxidation. Our conclusions show strong analogy with the model of Au Yang et al. (2019) who suggest $SO_2$ photo-oxidation on mineral dust could form sulfate aerosols depleted in $^{33}$S that would then be deposited. The residual $SO_2$ would be subsequently enriched in $^{33}$S, then be oxidized by common $O_3$, $H_2O_2$, $O_2$, OH oxidants. Their $^{33}$S-depletion

mechanism was not further constrained, except that it was speculated to be photochemical in origin.

If correct, this view requires reassessing the overall S-isotope fractionation during $SO_2$ atmospheric reaction. So far, previous studies assumed that the overall sulfur isotopic fractionation between the wet/dry deposit and oxidized $SO_2$ was equal to 1 (i.e. no isotope effect), but negative $\Delta^{33}$S in black crust is inconsistent with such an assumption.

Starting with $SO_2$ $\Delta^{33}$S-value of 0 ‰ (Au Yang et al., 2018; Lin et al., 2018b) and forming oxidized (sampled by secondary aerosols) and wet/dry deposit (sampled by black crusts) reservoirs with $\Delta^{33}$S-values up to 0.50 ‰ down to -0.34 ‰ respectively, mass balance imposes that $SO_2$ dry/wet depositions and secondary sulfate aerosols represents ~60 and 40 % respectively. This is in good agreement with proportions obtained by Chin et al. (2000) and reported by Harris et al. (2013b). Therefore, we conclude that MIE happening during $SO_2$ dry and wet

depositions could be a viable mechanism responsible for $^{33}$S-enrichment of secondary sulfate aerosols and that black crusts represent the $^{33}$S-negative complementary reservoir. Now, in order to better apprehend the aerosols/black crust complementarity, we modeled the S-isotopic fractionation of both black crusts and aerosols during $SO_2$ oxidation (Fig. 6 and 7). We assumed a Rayleigh distillation model to represent the atmosphere-



building stone interface open system. The global fractionation factor between residual $SO_2$ and oxidized

(secondary aerosols) + deposited (black crusts) $SO_2$ is defined as $\alpha_{global} = A\ \alpha_{BC-SO2}$ x B $\alpha_{aerosols-SO2}$ with A and B

are the proportions of $SO_2$ deposited and oxidized, being equal to 60 and 40 % respectively, which allow us to

deduce the $^{33,\ 34,\ 36}\alpha_{BC-SO2}$ and the associated $^{33,36}\beta$ factor. A $\delta^{34}S$ of 1 ‰ for the initial $SO_2$ was considered to

obtain black crusts of at least -3 ‰ (see section 5.1) and $\Delta^{33}S-\Delta^{36}S = 0$ ‰; $^{34}\alpha_{aerosols-SO2}$ was taken as 1.0097 as

calculated using the different oxidation channels proportions of Sofen et al. (2011). The oxidation being mass-

dependent, we chose $^{33,\ 36}\beta_{aerosols-SO2}$ of 0.515 and 1.9 respectively (Harris et al., 2012). The best fit is obtained for

$^{34}\alpha_{BC-SO2} = 0.9985$, $^{33}\alpha_{BC-SO2} = 0.9986$ and $^{36}\alpha_{BC-SO2} = 0.9972$ with $^{33,\ 36}\beta = 0.9$ and 1.9 respectively. The $^{33}S$-

enrichment in secondary sulfate aerosols is well represented by this parameterization (instantaneous and

cumulated products; Fig. 6 and 7). The concomitant $^{33}S$-depletion in modeled cumulated deposit is also well

represented but does not entirely capture black crusts isotopic compositions. The $^{33}S$-isotopic fractionation

occurring during the MIE is higher than the one observed in black crusts. At first order this model works,

predicting the total cumulated products of black crusts and aerosols to have $\Delta^{33}S$-values of -0.228 and 0.351

respectively. The match is not perfect, but remember that black crusts are produced from anthropogenic Parisian

$SO_2$ whereas aerosols formed in other locations possibly formed from distinct anthropogenic $SO_2$ $\delta^{34}S$-values. In

addition, we are aware that our model strongly depends on oxidation pathways estimated by Sofen et al. (2011),

which vary spatially and temporally but remain the most relevant so far. The main weakness is the too little

constrained estimate of intrinsic S-bearing compounds (CAS/plaster end-member) in the host-rock as well as

sulfate aerosols ($\Delta^{33}S > 0$ ‰) which dilutes the $^{33}S$ anomaly, lowering the overall black crust $\Delta^{33}S$. Ultimately,

black crusts result mainly from the deposition followed by oxidation of $SO_2$ on the building stone rather than

aerosols accumulation. The $\delta^{34}S$-value of initial (anthropogenic $SO_2$) is another little constrained parameter

whom variability might be difficult to estimate both spatially and temporally.

In conclusion, black crusts could represent the complementary sulfur end-member to sulfate aerosols. Its

fractionation factor is relatively restricted (-1.5 ‰) and is thus likely identifiable from its negative $\Delta^{33}S$-values.

Our model is actually consistent with assumption that the global $SO_2$ oxidation occurs with little fractionation

($^{34}\alpha_{global} = 1.00298$) as commonly done in the literature. Finally, the figure 8 summarizes the different sulfur

sources involved in the black crusts as well as the processes leading to their formation. Black crusts isotopic

compositions could then be explained by a mixing between sulfates from CAS/plaster ($\delta^{34}S \sim 18$ ‰ and $\Delta^{33}S = 0$

‰, see section 5.1 and 1 on Fig. 8), primary anthropogenic sulfates ($\delta^{34}S \sim -3$ ‰ and $\Delta^{33}S = 0$ ‰; see 2 on Fig. 8)

and wet/dry deposition of $SO_2$ undergoing MIE during its oxidation on the building stone combined with

secondary aerosols (see red triangle on Fig. 7 and 3 on Fig. 8).

## 6. Conclusion

Our study shows that black crusts do preserve an atmospheric signal of $SO_2$ oxidation, inferred from the non-zero

$\Delta^{17}O$. Part of the sulfate originates from the surrounding plaster and/or from the stone itself but overall > 60 %

originate from anthropogenic activities. We also discovered negative $\Delta^{33}S$ with near constant $\Delta^{36}S$ signatures,

which probably reflect magnetic isotope effect involving a new oxidation pathway. Magnetic isotope effect is

supposed to occur during the deposit of $SO_2$ on building stone surface (most likely carbonate), where $SO_2$ is



oxidized into sulfate leading to a [33]S-depletion in black crust sulfates. Therefore, the resulting [33]S-enrichment of residual $SO_2$ could account for positive $\Delta^{33}$S-values of sulfate aerosols observed worldwide, making black crust

sulfates their complementary $\Delta^{33}$S reservoir.

**Data availability**

All data needed to draw the conclusions in the present study re shown in this paper and/or the Supplement. For additional data related to this study, please contact the corresponding author (genot@ipgp.fr).


**Author contributions**

IG conducted oxygen isotope measurements under the supervision of EM and ELG at IPGP. DAY conducted sulfur isotope measurements at the GEOTOP (UQAM). IG and EM collected the samples. IG, PC, EM and DAY interpreted the data. IG wrote the paper with contributions from all coauthors. EM and MR conceived the project.


**Competing interests**

The authors declare that they have no conflict of interest.

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



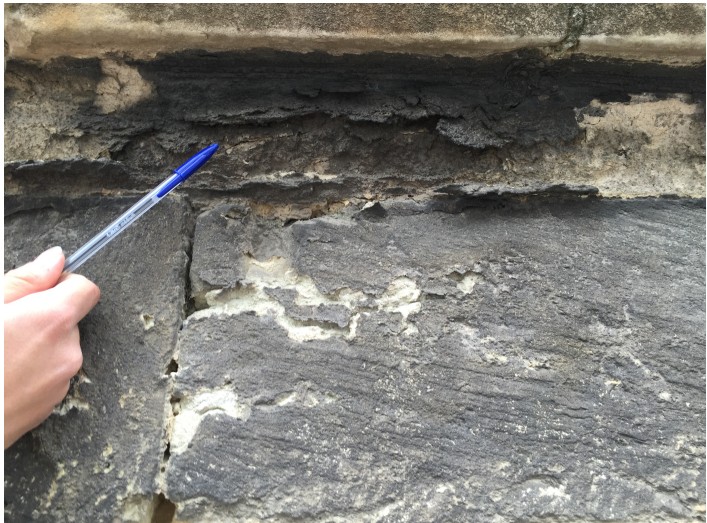


**Fig. 1** Thin layer of black crusts formed on a carbonate building stone, on a church wall in Fécamp city.

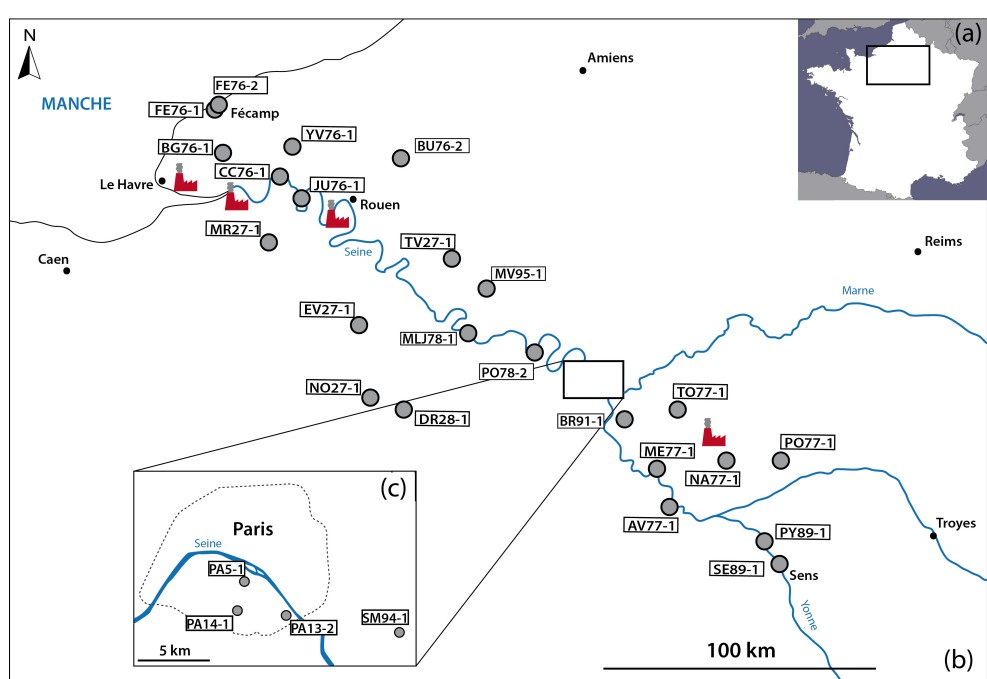


**Fig. 2** Maps with the sampling location. **a** Location of the studied area in the Northern Paris Basin on the map of France. **b** The NW-SE cross-section from Fécamp to Sens with the 27 samples and the four power plants (in red). **c** Focus on the samples located in the Paris area.





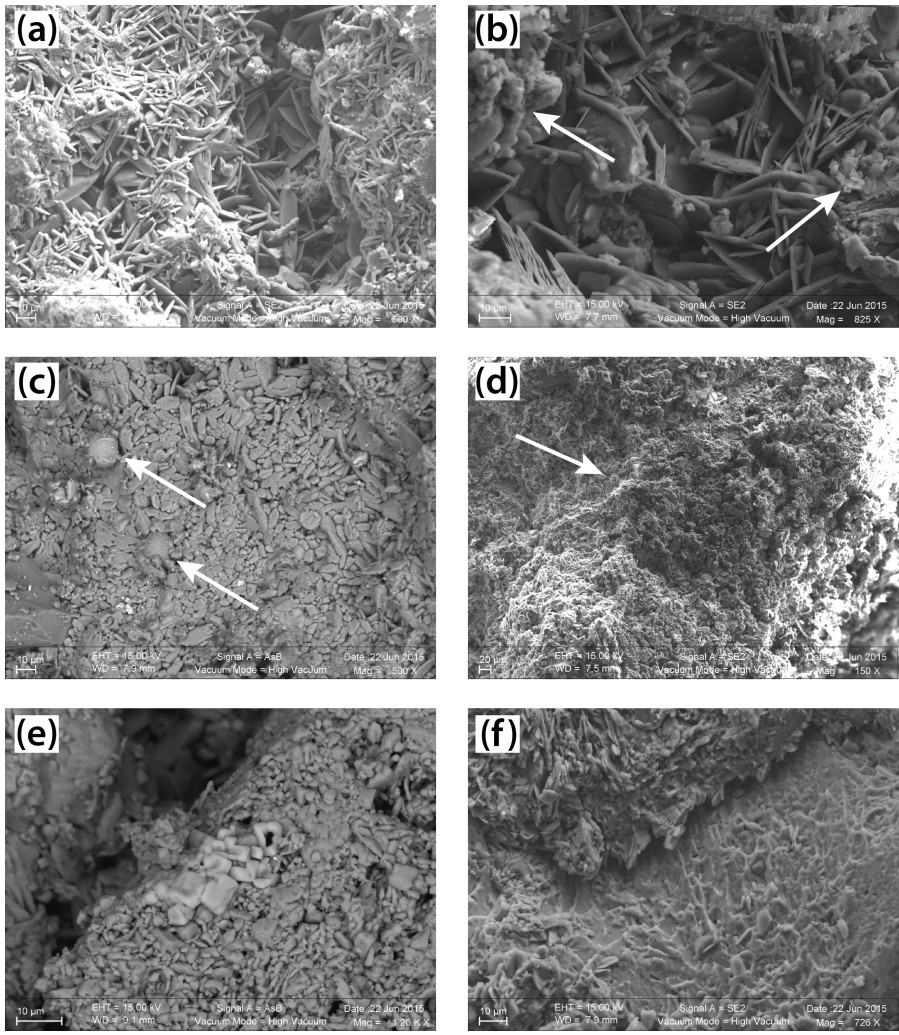

**Fig. 3** SEM images of black crusts samples from Paris (PA14-1, PA13-2) and Montfort-sur-Risles (MR27-1). **a** Two distinct layers into the crust: an upper opaque one with aggregates of particulate matter and clay minerals (left and right sides of the picture) and a more crystallized one with acicular gypsum crystals perpendicular to the host substrate. **b** Presence of soot (arrow) on the two layers (PA14-1). **c** Fly ashes (arrow) with the formation of small gypsum crystals on their surfaces (PA13-2). **d** Large amount of fly ashes (arrow and smaller not indicated) and soot in MR27-1 sample, located in a rural place. **e** Isolated cubic crystals of halite (NaCl) in MR27-1 sample, at 28 km from the coastline. **f** Dissolution of the underlying limestone (on the bottom) and subsequently the precipitation of gypsum (on the top).

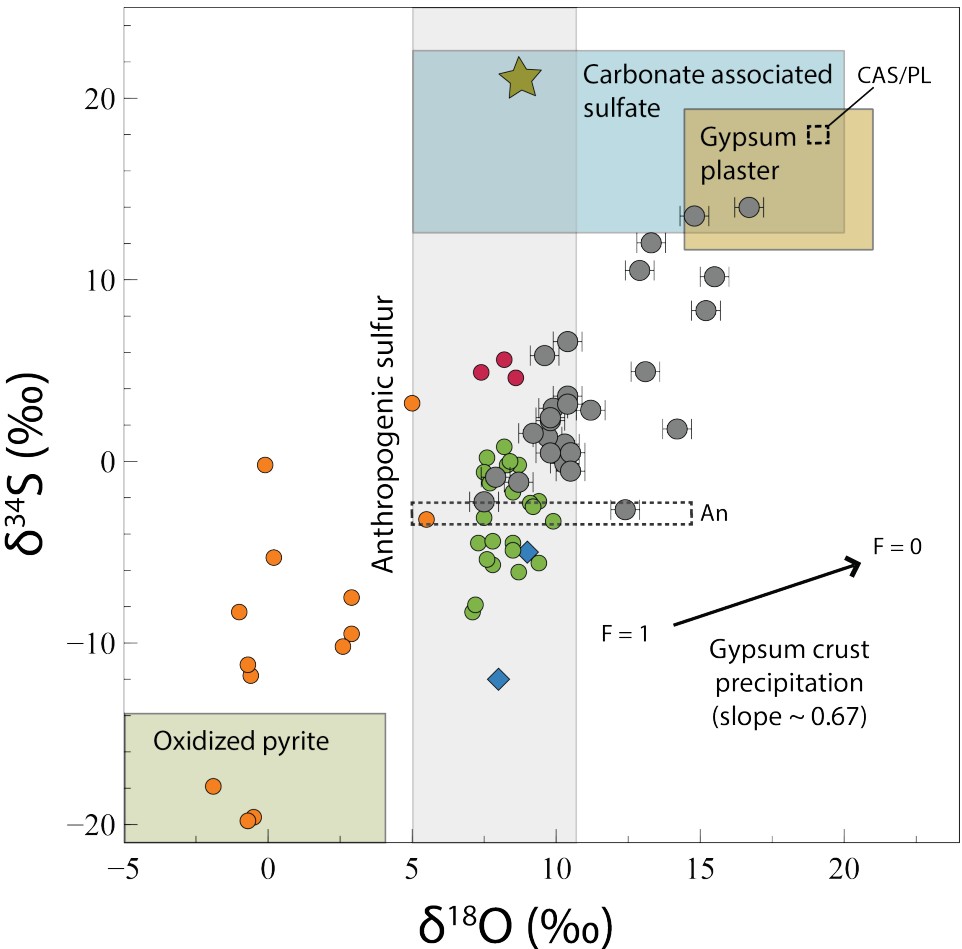

**Fig. 4** Evolution of $\delta^{34}S$ in function of $\delta^{18}O$ in black crusts sulfates. Grey points are from this study, green and red points represent the isotopic compositions of black crusts from Antwerp (Torfs et al., 1997) and Venice (Longinelli and Bartelloni, 1978) respectively. Orange and blue points represent the isotopic compositions of black crusts from Ljubljana (Kramar et al., 2011) and Bourges (Vallet et al., 2006) respectively, probing potentially an oxidized pyrite source. The yellow star represents the modern seawater (Markovic et al., 2016; Rees et al., 1978). The extreme anthropogenic sulfate isotopic compositions are from Lee et al. (2002) for oxygen and from Faure (1986) for sulfur. The carbonate-associated sulfates compositions are from (Rennie and Turchyn, 2014; Turchyn et al., 2009) and those of gypsum plaster come from (Kloppmann et al., 2011). Isotopic compositions determining the oxidized pyrite field is from Canfield (2004) for sulfur and is calculated following a Rayleigh distillation model with an initial $H_2O$ $\delta^{18}O$ = 6 ‰ and a mean fractionation factor of 1.010 (Gomes and Johnston, 2017) for oxygen. The black arrow represents the fractionation induced by gypsum precipitation where F=1 mean that all sulfates are dissolved and F=0 means that all sulfates are precipitated. The dashed fields represent the sulfur isotopic composition of the two anthropogenic (An) and CAS/plaster (CAS/PL) end-members.

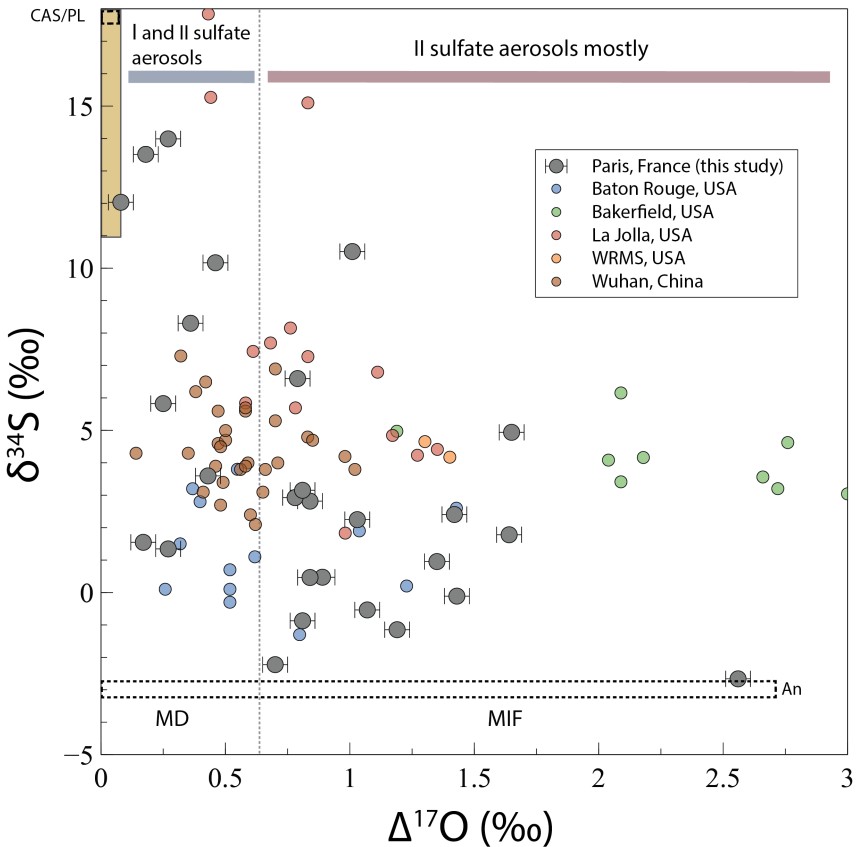

**Fig. 5** Evolution of $\delta^{34}S$ in function of $\Delta^{17}O$ in black crusts sulfates. The limit between mass-dependent and mass-independent fractionation (dashed line) is defined for $\Delta^{17}O$ ~0.65 ‰, where $H_2O_2$ will be the major oxidant, giving its O-anomaly to sulfates (Savarino et al., 2000). When $\Delta^{17}O$ < 0.65 ‰, black crusts sulfates result from a mixing between primary sulfates (gypsum plaster and CAS and/or anthropogenic sulfur) and secondary sulfate aerosols where $SO_2$ is oxidized by $H_2O_2$ ($\Delta^{17}O$ = 0.65 ‰), OH ($\Delta^{17}O$ = 0 ‰) or $O_2$-TMI ($\Delta^{17}O$ = -0.09 ‰) mainly

(grey bar). When $\Delta^{17}O$ > 0.65 ‰, black crusts sulfates represent secondary sulfate aerosols mainly, resulting from a mixing between $SO_2$ oxidized by $O_3$ ($\Delta^{17}O$ = 8.75 ‰) and $H_2O_2$ (red bar). The yellow array represents gypsum plaster and CAS isotopic compositions. The data of Baton Rouge are from Jenkins and Bao (2006), those of Bakerfield, La Jolla and WMRS (White Mountain Research Station) are from Lee and Thiemens (2001); Romero and Thiemens (2003) and those of Wuhan are from (Li et al., 2013). The dashed fields represent the sulfur

isotopic composition of the two anthropogenic (An) and CAS/plaster (CAS/PL) end-members.






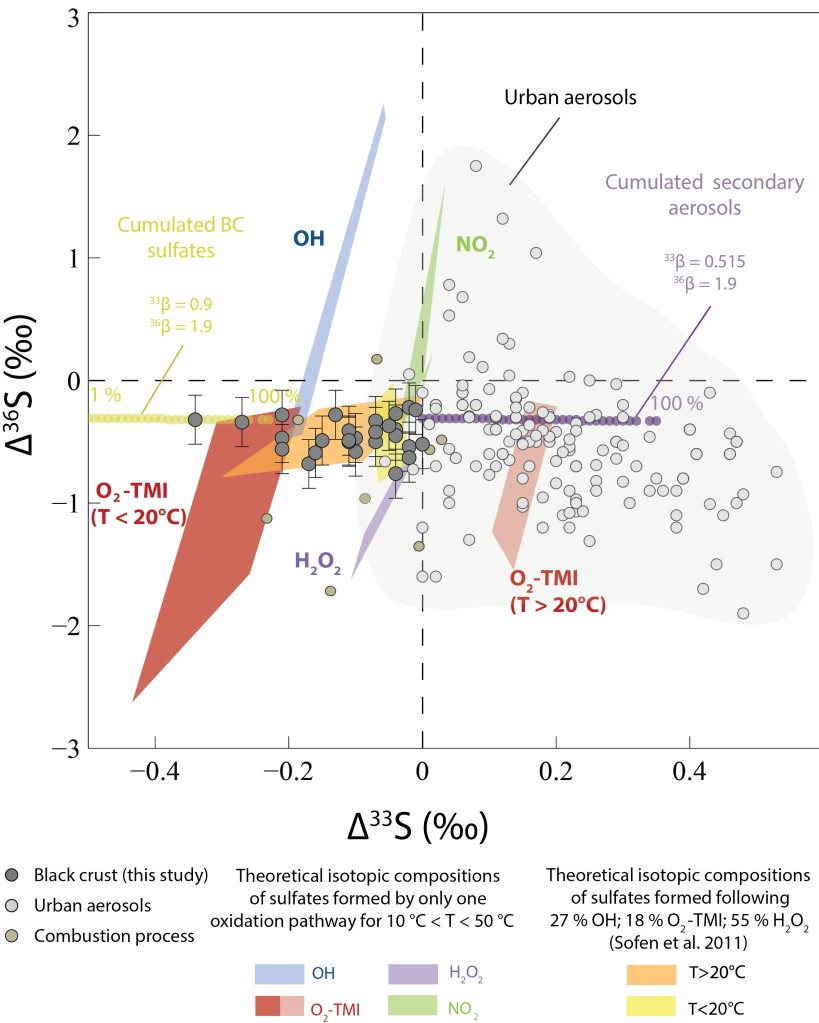

**Fig. 6** $\Delta^{33}S$ and $\Delta^{36}S$ of the black crust, compared to sulfates formed by different oxidation pathways and by a mixing of them in the proportions estimated by Sofen et al. (2011). We took $^{33}\beta_{H2O2/O3} = 0.511$, $^{33}\beta_{OH} = 0.503$, $^{33}\beta_{O2-TMI} = 0.498$ (for T < 20°C), $^{33}\beta_{O2-TMI} = 0.547$ (for T > 20°C), $^{33}\beta_{NO2} = 0.514$ and $^{36}\beta_{H2O2/O3} = 1.82$, $^{36}\beta_{OH} = 1.97$, $^{36}\beta_{O2-TMI} = 1.98$ (for T < 20°C and T > 20°C), and $^{36}\beta_{NO2} = 1.90$ (Au Yang et al., 2018; Harris et al., 2013b). As $^{33}\beta_{O3}$ and $^{36}\beta_{O3}$ are unknown, we modified the proportions of Sofen et al. (2011) as follows: 27 % OH, 18 % $O_2$-TMI, 55 % $H_2O_2$ and 0 % $O_3$. The urban aerosols isotopic compositions are a compilation from Au Yang et al. (2019); Guo et al. (2010); (Lin et al., 2018b); Romero and Thiemens (2003); Shaheen et al. (2014) while the combustion process reflect samples from (Lee et al., 2002). Modeled $\Delta^{33}S$-$\Delta^{36}S$-values of cumulated black crusts (BC) sulfates formed by $SO_2$ wet/dry deposition with a MIE ($^{33}S$-depletion compared to initial $SO_2$ with constant negative $\Delta^{36}S$) and of cumulated secondary aerosols formed by $SO_2$ oxidation by $O_3$, $O_2$-TMI, OH, $H_2O_2$ ($^{33}S$-enrichment compared to initial $SO_2$) from an initial $SO_2$ with $\Delta^{33}S$-$\Delta^{36}S$ = 0 ‰ are reported with corresponding β exponents (see section 5.3.2 for model explanation). Residual $SO_2$ and global cumulated BC + secondary aerosols isotopic compositions were not reported for better readability. Percentages indicate the fraction of produced cumulated BC and secondary aerosols.




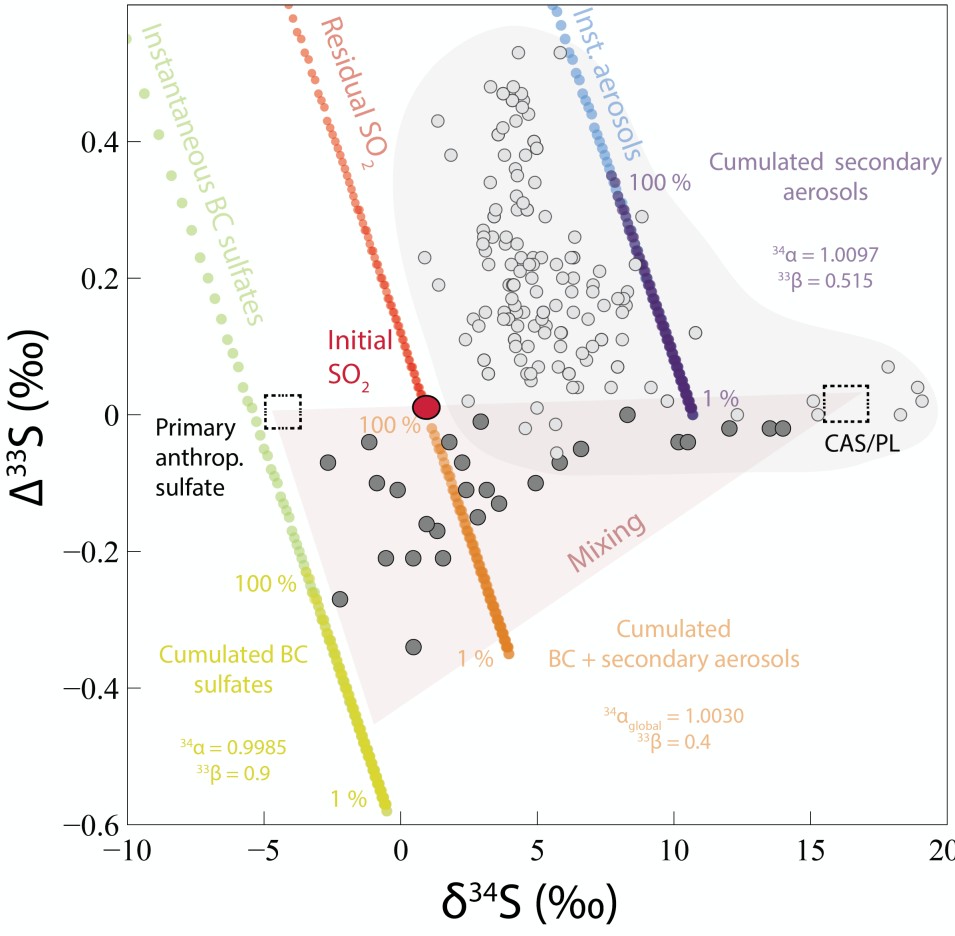

**Fig. 7 a** Modeled $\delta^{34}S$ and $\Delta^{33}S$-values of black crusts (BC) sulfates (instantaneous and cumulated) formed by $SO_2$ wet/dry deposition with a MIE ($^{33}S$-depletion compared to initial $SO_2$) and of secondary aerosols (instantaneous and cumulated) formed by $SO_2$ oxidation by $O_3$, $O_2$-TMI, OH, $H_2O_2$ ($^{33}S$-enrichment compared to initial $SO_2$) from an initial $SO_2$ with $\delta^{34}S = 1$ ‰ and $\Delta^{33}S = 0$ ‰ (red point). $^{33}S$-enrichment of residual $SO_2$ and global cumulated BC + secondary aerosols isotopic compositions are also reported. Percentages indicate the fraction of produced cumulated BC and secondary aerosols (see section 5.3.2 for model explanation). Black crust sulfate isotopic compositions (dark grey points) can be explained by a mixing (red triangle) between sulfates from CAS/plaster (dashed square, $\delta^{34}S = 18$ ‰ and $\Delta^{33}S = 0$ ‰, see section 5.1), primary anthropogenic sulfates (dashed square, $\delta^{34}S = -3$ ‰ and $\Delta^{33}S = 0$ ‰) and sulfates formed by wet/dry deposition of $SO_2$ undergoing a MIE and oxidized $SO_2$ forming secondary aerosols (cumulated BC sulfates and cumulated BC + secondary aerosols). Urban aerosols (light grey points) isotopic compositions are a compilation from Au Yang et al. (2019); Guo et al. (2010); (Lin et al., 2018b); Romero and Thiemens (2003); Shaheen et al. (2014).



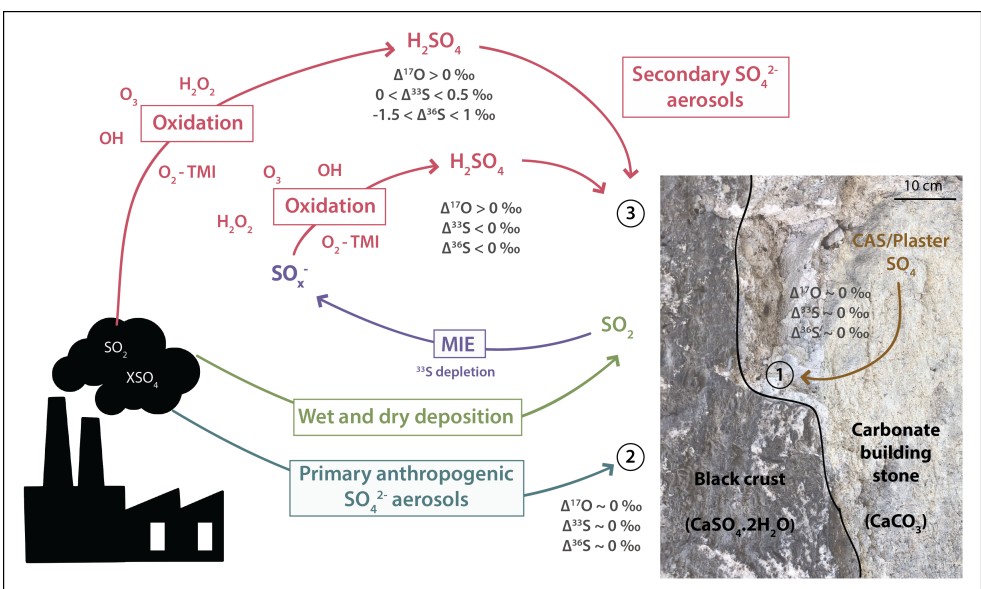


**Fig. 8** Scheme summarizing the sulfur sources and processes that lead to black crusts formation. Sulfur dioxide releases by anthropogenic activities can either be oxidized in the atmosphere by $H_2O_2$, $O_3$, OH, $O_2$-TMI and formed secondary sulfate aerosols that will react with the carbonate building stone to produce [33]S-enriched black crusts sulfates or be deposited, as dry/wet deposit, on the carbonate substrate where its oxidation into $SO_x^-$ then sulfates through MIE will produce [33]S-depleted black crusts sulfates and a [33]S-enriched residual $SO_2$ (source 3). Primary sulfates emitted by anthropogenic activities (source 2) or carbonate-associated sulfates and/or plaster of the host-rock (source 1) are also likely sources contributing to black crusts formation.








| Samples | Location | Orientation of sampled faces | Distance from the sea (km) | Height above the ground (m) | Exposition to traffic road |
|---|---|---|---|---|---|
| PA14-1 | 48° 49' 37.97" N 2° 20' 5.21" E | 65° N | 170 | 1.5 - 2.0 | Directly exposed |
| PA13-2 | 48° 49' 26.42" N 2° 22' 33.48" E | 150° N | 170 | 2.0 | Directly exposed |
| PA5-1 | 48° 50' 37.09" N 2° 20' 25.77" E | 14° N | 170 | 1.5 | Directly exposed |
| BR91-1 | 48° 42' 11.15" N 2° 30' 28.82" E | 107° N | 190 | 1.5 | Directly exposed |
| PO78-2 | 48° 55' 42.41" N 2° 2' 16.94'' E | 100° N | 150 | 1.5 - 2.0 | Directly exposed |
| MLJ78-1 | 48° 59' 32.40" N 1° 42' 31.78" E | 295° N | 135 | 1.2 - 1.5 | Directly exposed |
| SM94-1 | 48° 48' 47.00" N 2° 28' 28.84" E | 0° N | 175 | 2.0 - 3.5 | Directly exposed |
| TO77-1 | 48° 44' 18.74" N 2° 46' 7.78" E | 21° N | 202 | 1.5 – 2.0 | Directly exposed |
| MV95-1 | 49° 9' 3.57" N 1° 47' 13.53" E | 343° N | 110 | 1.5 | Not directly exposed |
| TV27-1 | 49° 14' 8.17" N 1° 36' 27.30" E | 340° N | 95 | 1.5 – 2.0 | Directly exposed |
| BU76-2 | 49° 35' 2.94" N 1° 21' 23.79" E | 313° N | 45 | 1.5 – 2.0 | Directly exposed |
| YV76-1 | 49° 37' 1.12" N 0° 45' 16.36" E | 275° N | 28 | 1.5 – 2.0 | Not directly exposed |
| FE76-1 | 49° 45' 31.47" N 0° 22' 2.74" E | 190° N | 0,5 | 1.5 – 2.0 | Directly exposed |
| FE76-2 | 49° 45' 29.26" N 0° 22' 35.97" E | 0° N | 1,1 | 1.5 – 2.0 | Not directly exposed |
| BG76-1 | 49° 35' 30.11" N 0° 25' 39.22" E | 151° N | 21 | 1.5 | Not directly exposed |
| CC76-1 | 49° 31' 32.55" N 0° 43' 50.12" E | 191° N | 37 | < 2.0 | Directly exposed |
| JU76-1 | 49° 25' 56.69" N 0° 49' 8.42" E | 317° N | 50 | 1.5 – 2.0 | Directly exposed |
| MR27-1 | 49° 17' 40.51" N 0° 39' 52.77" E | 120° N | 28 | < 2.0 | Directly exposed |
| EV27-1 | 49° 1' 25.82" N 1° 8' 29.45" E | 158° N | 84 | 2.0 | Not directly exposed |
| NO27-1 | 48° 46' 15.78" N 1° 11' 53.81" E | 13° N | 103 | 1.5 | Not directly exposed |
| DR28-1 | 48° 44' 9.39" N 1° 22' 5.06" E | 233° N | 115 | 1.5 – 2.0 | Not directly exposed |
| ME77-2 | 48° 32' 20'' N 2° 39' 33 '' E | 22° N | 210 | 1.3 | Not directly exposed |



| | | | | | |
|---|---|---|---|---|---|
| AV77-1 | 48° 24' 15 '' N<br>2° 43' 2'' E | 73° N | 230 | 1.3 – 2.3 | Directly exposed |
| SE89-1 | 48° 12' 8'' N<br>3° 16' 24'' E | 263° N | 270 | 1.7 | Not directly exposed |
| PY89-1 | 48° 17' 16'' N<br>3° 12' 16'' E | 343° N | 263 | 1.5 – 2.1 | Not directly exposed |
| PO77-1 | 48° 33' 38'' N<br>3° 17' 29'' E | 23° N | 230 | 1.5 – 2.0 | Not directly exposed |
| NA77-1 | 48° 33' 26'' N<br>3° 0' 25'' E | 351° N | 230 | 2.5 | Not directly exposed |

**Table 1** Characteristics of black crusts samples. Their name was given according to the city and the department where they are located and following the number of samples gathered at the same place (NA77-1: NA = Nangis; 77 = department; -1 = first sample collected).





055

| (2σ) | $\delta^{18}O$<br>± 0.5 ‰ | $\Delta^{17}O$<br>± 0.05 ‰ | $\delta^{34}S$<br>± 0.2 ‰ | $\Delta^{33}S$<br>± 0.01 ‰ | $\Delta^{36}S$<br>± 0.2 ‰ | Distance from<br>coastline (km) |
|---|---|---|---|---|---|---|
| PO78-2 | 9.7 | 0.27 | 1.3 | -0.17 | -0.68 | 150 |
| AV77-1 | 9.8 | 1.03 | 2.2 | -0.07 | -0.33 | 230 |
| PY89-1 | 9.8 | 0.84 | 0.5 | -0.21 | -0.47 | 263 |
| PO77-1 | 12.4 | 2.56 | -2.7 | -0.07 | -0.50 | 231 |
| BG76-1 | 13.1 | 1.65 | 5.0 | -0.10 | -0.47 | 21 |
| MV95-1 | 15.5 | 0.46 | 10.2 | -0.04 | -0.40 | 110 |
| BR91-1 | 14.8 | 0.18 | 13.5 | -0.02 | -0.22 | 190 |
| JU76-1 | 9.9 | 0.78 | 3.0 | -0.01 | -0.24 | 50 |
| TV27-1 | 16.7 | 0.27 | 14.0 | -0.02 | -0.54 | 95 |
| YV76-1 | 12.9 | 1.01 | 10.5 | -0.04 | -0.27 | 28 |
| DR28-1 | 7.5 | 0.70 | -2.2 | -0.30 | -0.34 | 115 |
| FE76-1 | 10.3 | 1.35 | 0.9 | -0.16 | -0.59 | 0.5 |
| NO27-1 | 14.2 | 1.64 | 1.8 | -0.04 | -0.45 | 103 |
| ME77-2 | 10.5 | 1.07 | -0.5 | -0.21 | -0.29 | 210 |
| PA13-2 | 7.9 | 0.81 | -0.9 | -0.10 | -0.58 | 170 |
| MLJ78-1 | 15.2 | 0.36 | 8.3 | 0.00 | -0.52 | 135 |
| EV27-1 | 10.4 | 0.79 | 6.6 | -0.05 | -0.37 | 84 |
| FE76-2 | 8.7 | 1.19 | -1.1 | -0.04 | -0.76 | 1.1 |
| SE89-1 | 10.4 | 0.81 | 3.2 | -0.11 | -0.41 | 270 |
| BU76-2 | 10.3 | 1.43 | -0.1 | -0.11 | -0.50 | 45 |
| MR27-1 | 11.2 | 0.84 | 2.8 | -0.15 | -0.49 | 28 |
| PA14-1 | 9.2 | 0.17 | 1.5 | -0.21 | -0.56 | 171 |
| TO77-1 | 13.3 | 0.08 | 12.0 | -0.02 | -0.64 | 202 |
| NA77-1 | 9.8 | 1.42 | 2.4 | -0.11 | -0.49 | 232 |
| PA5-1 | 10.5 | 0.89 | 0.5 | -0.34 | -0.32 | 172 |
| SM94-1 | 10.4 | 0.43 | 3.6 | -0.13 | -0.28 | 175 |
| CC76-1 | 9.6 | 0.25 | 5.8 | -0.07 | -0.43 | 37 |

**Table 2** $\delta^{18}O$, $\delta^{34}S$, $\Delta^{17}O$, $\Delta^{33}S$ and $\Delta^{36}S$ measures of each sample with the distance from coastline.