# Peer review of "Oxygen and sulfur mass-independent isotopic signatures in black crusts: the complementary negative $\Delta^{33}$ S-reservoir of sulfate aerosols?"

_Atmospheric Chemistry and Physics, 2019_

## Referee Comment (RC1) · Anonymous Referee #1 · 20 Nov 2019

Black crusts contain sulfate of atmospheric deposition integrated over a period of time. They are an archive of atmospheric sulfate, some of which are secondary atmospheric sulfate derived from the oxidation of SO2 (mostly). This study sampled a set of black crusts among Paris metropolis and discovered that many of the sulfate samples bear negative $\triangle$33S values while nearly invariable in the $\triangle$36S. This prompted the authors to conclude that black crusts or associated heterogeneous oxidation of SO2 may have been the "missing" pool of the $\triangle$33S-negative sulfur being sought to balance the observed, largely $\triangle$33S-positive atmospheric sulfate aerosols. The authors also specu-

lated the mechanism to be a magnetic isotope effect associated with heterogeneous radical reactions. This is an interesting discovery that may provide clues to the mysterious 33S anomalies in modern atmospheric sulfate. However, the current draft could benefit from some major revisions before being accepted for publication. 1. Some of the discussion parts are unnecessarily lengthy especially considering the insignificance of the problems in question. The proportion of natural vs. anthropogenic estimates have been done before and usually bear a large uncertainty and is not a critical problem. Those lengthy discussions and estimates are diluting the important discoveries in this study. I suggest trying to trim the text down to 50% of the current length. Focus on new things, the $\Delta$33S, and the $\Delta$33S and $\Delta$36S correlation. 2. I could guess from the text that at least two writers were writing this manuscript. Make sure the English and flow are consistent. 3. Some of the specifics listed below are syntactic and some are conceptual and should be dealt with in diligence. Line 26-29: This is inconsistent with the many published negative $\Delta$33S data from Beijing, e.g. Han et al., 2017. Line 33: Not necessarily going through a H2SO4 phase; "they" should be "that". Line 38: "influent" is not a good word here; delete "gases". Line 46: not necessarily "distant". Line 51: To many, there are three pathways: gas (homogeneous), aqueous, and heterogeneous (surface). Line 54: characterizing Line 81: "Intrinsic" is a poor choice here. Line 88-89: Few sulfate samples have been measured for all the 4 sulfur and 3 oxygen isotope compositions together. Thus, this is not a significant thing to say. Line 105-114: There are numerous conceptual misunderstanding and inaccuracies in these writings. I suggest delete them all. Line 114-115: Some of the deviations maybe still be mass-dependent under this definition per se. Line 117: I suggest you use 0.5305 for the sake of internal consistency. Note that both 0.515 and 1.889 are the high-temperature limit values for quadruple sulfur isotope system. For triple oxygen isotope system at high-T limit, the exponent value is 0.5305. Line 158: We had a similar correction factor. This correction will have +/-2‰ error (1 sigma). Sample impurity and therefore O2 yield has been the major source of errors. Thus, the actual error for $\delta$18O of sulfate could be much larger. Line 160: Change "during" to "for". Line 162: precipitated as Line

185: Being consistent with … Line 193: delete words after and including "highlight", partly because "anthropogenic emissions" is poorly defined. Line 197: change "can" to "may". Line 215: delete ",". Line 222: etc. Line 225: "extrinsic" and "intrinsic" are not ideal words here. Line 228-239: These discussions are not necessary because the Rayleigh process requires sampling from the residues or products of the same reservoir during evolution and the black crust gypsum is not. Line 240-252: I think this is because Harris et al (2012)'s fractionation factors are not for multiple steps with multiple oxygen sources and are only applicable in their particular experimental settings. Line 256, 257, 261 ...: significant digits should reflect experimental error, in the case of $\delta$34S, it should be at most at the second decimal points. Line 297-298: Check the English Line 322: you meant "between a less variable" instead? Line 334-335: A very confusing sentence. Line 372: Delete "implying a lower pCO2 and/or a higher flux of sulfide re-oxidation in sediments". It's a distraction. Line 393-402: This O3-H2O2 proportion exercise is not only too simplified but also invalid because you did not consider the contribution of Fe-Mn catalyzed oxidation by O2 in aqueous condition, a pathway that is known to be significant. Line 444-445: The sentence "resulting in negative $\Delta$33S-$\Delta$36S but not low enough to explain $\Delta$33S < -0.2 ‰'" is ambiguous here. Line 483: "than", not "that". Line 521-523: I'd rather see this "microbial ..." sentence deleted entirely. Line 564: Change "little" to "poorly". Line 565: change to "whose". Line 570: delete "the" before "figure 8".
* * *

---

## Referee Comment (RC2) · Mang Lin (Referee) · 31 Dec 2019

Small but significant mass-independent fractionation anomalies of quadruple sulfur isotopes (S-MIF) have been widely observed in today's atmospheric sulfate aerosols. Similar isotopic signatures were also observed in ice and lake sediment records. These observations were somehow unexpected in that they are in contrast with the traditional assumption that S-MIF in sulfate aerosols are ONLY produced in the stratosphere by UV-induced photochemical reactions of SO2. The fundamental processes leading to such "unexpected" S-MIF signature are being highly debated recently and several

mechanisms have been proposed. More measurements in the modern atmosphere and in geological records containing deposited atmospheric sulfur are important for testing existing predictions and for a more precise understanding of S-MIF chemical physics and atmospheric sulfur cycle.

In this manuscript, the authors present a new set of quadruple sulfur isotope data in gypsum-forming "black crusts". Measurements of triple oxygen isotopes in these sulfate samples suggested that most sulfates samples are of atmospheric origins. An interesting finding is that most quadruple sulfur isotope data are characterized by negative D33S values (from -0.34 per mil to ∼0 per mil), a pattern different from most observation in the past. The variation of D36S values (from -0.7 per mil to -0.2 per mil) is however relatively small compared to previous studies. The authors interpret these observations as a signature of magnetic isotope effects during SO2 oxidation on carbonate, although the effect was not tested by laboratory experiments and a detailed mechanism was not proposed. They then argue that such reaction will lead to positive D33S values in residual SO2, which explain the positive D33S values in most atmospheric sulfates. Overall, the isotope data are novel. The interpretation, which is subject to further discussion and validation, is testable. This work is deserved to be published in ACP. I have several suggestions as follows to improve the manuscript before its final publication.

1. The manuscript is too long and there is too much information. I think the most important finding of this manuscript is the observation of negative D33S values. I believe that the authors agree with me as they also put this information in the title. However, the authors used 5-6 pages to discuss d18O, d34S, and D17O data, which may be distracting. I agree that some discussions in those parts are important to support the authors' interpretation, but in my point of view, many of them may be unnecessary and even irrelevant (e.g., lines 370-375). I understand that everyone has his/her own writing style, but I am afraid that some readers will get lost in this manuscript. My suggestion is that the authors should shorten the d18O, d34S, and D17O discussions, and highlight

their most important findings (i.e., the negative D33S data). Though I will leave it for the authors to decide as to whether they prefer keeping their writing style.

2. In the introduction, the current state and gap of our knowledge on S-MIF (especially D33S) is not well presented. It is true that most aerosol measurements displayed positive D33S values, but negative D33S values were also noted by previous works by Lee et al. (2002), Shaheen et al. (2014), Han et al. (2017), and Lin et al. (2018a). In addition, I suggested the authors highlighted their recent work (Au Yang et al., 2019) in the introduction. Au Yang et al., (2019) suspected that photooxidation of SO2 on the surface of mineral dusts may produce large S-MIF signatures, and the gypsum layer on carbonate studied in this study is a natural laboratory to test their hypothesis and concept model.

3. The authors listed some end-members to quantify the anthropogenic contribution, but it is not clear how the authors defined their isotopic values. For example, in line 289, it was mentioned that the end-member CAS/PL has a D17O value of zero per mil. Did the author measure the CAS/PL? Is this value just a simple predication? In line 291, it was mentioned that the other end-member possesses d18O values ranging from 5 to 15 per mil. I am not sure where the authors obtained these numbers. Can the authors clarify? The definition of anthropogenic emission endmember (d34S = -3 per mil) is also unclear and may be problematic. It is well-known that d34S values of anthropogenic emitted sulfur are highly variable. This fact is also noted by the authors in lines 314-316. Given that the 34S value could be from -30 per mil to 30 per mil as cited by the authors, I am confused why the value of -3 per mil was selected. I checked Montana et al. (2012) cited by the authors but cannot find the value of -3 per mil. I suggest the authors to check previous d34S measurements of SO2 and sulfate in the same studied region, if there is any, and use those data as end-members. The uncertainties should be also considered and briefly discussed.

4. I would like to point out a possibility that pyrite could be oxidized by H2O2 or even O3. This is shown by a recent talk in the AGU fall meeting this month

(https://agu.confex.com/agu/fm19/meetingapp.cgi/Paper/597703). I am not saying that such oxidation pathways play a dominant role in the authors' samples, but the authors should carefully consider this possibility. One can also argue that the authors' samples are mixed by three end-members (CAS/PL, anthropogenic sulfur, and oxidized pyrites).

5. The D17O calculation in line 397 is wrong. The measured D17O value is also controlled by other oxidants such as OH radical and O2. In the calculation done by the authors, the estimated contribution of S(IV)+O3 oxidation pathway is just the lower limit as the authors assume the S(IV)+H2O2 reaction is the only other oxidation pathway. The authors can also estimate the maximum contribution of S(IV)+O3 oxidation pathway by assuming no contribution from H2O2.

6. Line 514: This is a misleading statement. It is an "interpretation" instead of "observation". Non-zero D33S value is still a prediction and has not been confirmed by any measurement yet. It is important to do such measurements in the future to test the authors' prediction though.

7. In the discussion of magnetic isotope effects, the authors suggested that they cannot rule out the effect of micro-organisms. If isotopic compositions in black crusts were linked to magnetic isotope effects from microbial activities, how do the authors explain the D17O data?

8. I suggest putting all legends on all figures. It is difficult to check the caption one by one.

---

## Author Comment (AC1) · 18 Feb 2020

First of all, there was no rebuttal to our interpretations and the comments raised by the first referee were important to improved our manuscript. We almost followed all of them.

1) "Some of the discussion parts are unnecessarily lengthy especially considering the insignificance of the problems in question. The proportion of natural vs. anthropogenic estimates have been done before and usually bear a large uncertainty and is not a

critical problem. Those lengthy discussions and estimates are diluting the important discoveries in this study. I suggest trying to trim the text down to 50% of the current length. Focus on new things, the $\Delta 33S$, and the $\Delta 33S$ and $\Delta 36S$ correlation."

We think that before addressing if $\Delta 33S$ anomaly lies in a "process" rather than a "source", it is important to check if S and O isotopic compositions are affected by processes (e.g. SO2 partial oxidation, gypsum precipitation...) even if it only slightly overprint the source signatures. It would otherwise confuse the reader having two sections, one dealing with mixing (d34S-d18O-$\Delta 17O$), the other dealing with processes (d34S-$\Delta 33S$-$\Delta 36S$). We added our developed reasoning L247-251. Given that d34S-d18O variation is explained by a mixing, representing source signatures and that $\Delta 17O$-values are evidence for atmospheric aerosols, we know that $\Delta 33S$ signature is not overprinted and that the source or process leading to this anomaly is atmospheric (not coming from the host-rock). We merged d34S-d18O-$\Delta 17O$ sections, keeping the discussion about processes even if mixing is the main mechanism and reduced the text by $\sim$ 1.5 pages for this part of the discussion, almost 20% of the 50% asked by the first referee, whereas the second referee leaves us the decision to reduce or not. Furthermore, we added details asked by both referees which does not allow us to reduce the text as much. The discussion is very detailed because we think that most of the time, the interpretation is limited to a mixing without discussing other processes. Furthermore, various scientific communities could be interested in this paper without understanding all concepts covered (e.g. black crusts, stable isotope geochemistry in the atmosphere, mass-independent fractionation). Consequently, we wish to keep a detailed, rigorous and accordingly developed discussion to allow everyone a good comprehension.

2) "I could guess from the text that at least two writers were writing this manuscript. Make sure the English and flow are consistent."

We agree and tried to homogenize the English in the manuscript.

3) We took into account all the syntactic comments suggested by the referee, not listed here.

4) "Line 26-29: This is inconsistent with the many published negative $\Delta33S$ data from Beijing, e.g. Han et al., 2017."

We clarified this aspect, adding L27 that "except for a few samples", sulfate aerosols have mostly positive $\Delta33S$.

5) For the following comments: "Not necessarily going through a H2SO4 phase" we added "can" L34. "Influent" is replaced by "efficient particles" L39. "Not necessarily distant" we added "a local or more distant source" L46-47. For the oxidants, we used the simplification suggested by the referee that is the three main pathways L47-48.

6) "Line 81: "Intrinsic" is a poor choice here. and Line 225: "extrinsic" and "intrinsic" are not ideal words here."

We agree but we only reported this term because used in the literature by Kramar et al., (2011) and Kloppmann et al., (2011). Because many distinctions exist for sulfate aerosols (primary/secondary; natural/anthropogenic...), we removed the intrinsic/extrinsic distinction in the discussion, only recalling its use in the literature L85.

7) "Line 88-89: Few sulfate samples have been measured for all the 4 sulfur and 3 oxygen isotope compositions together. Thus, this is not a significant thing to say."

We wanted to highlight that this is the first time for black crusts samples. We agree it is not the case for sulfate aerosols and we removed this emphasis L91-95.

8) "Line 105-114: There are numerous conceptual misunderstanding and inaccuracies in these writings. I suggest delete them all. Line 114-115: Some of the deviations maybe still be massdependent under this definition per se. Line 117: I suggest you use 0.5305 for the sake of internal consistency. Note that both 0.515 and 1.889 are the high-temperature limit values for quadruple sulfur isotope system. For triple oxygen isotope system at high-T limit, the exponent value is 0.5305."

We explained in another way, L110-121, the concept of mass-dependent vs mass-independent fractionation depending on how large is the deviation from the mass-dependent curve, without using the term "anomaly" which was indeed ambiguous. We also applied 0.5305 for 17 (L104, L108 and Eq. 1), specifying that all the used here are the high temperature limit (L108), as suggested by the referee.

9) "Line 158: We had a similar correction factor. This correction will have +/-2‰ error (1 sigma). Sample impurity and therefore O2 yield has been the major source of errors. Thus, the actual error for d18O of sulfate could be much larger."

We believe that our correction factor can be applied to all samples without such a significant error because 1) all samples were purified on an ion-exchange resin to get only the sulfates (protocol implemented by Legendre et al., 2017) and 2) the reproducibility between duplicate samples is $\sim$ 0.3‰ lower than the NBS127 reproducibility. If the O2 yield was different for one sample, we should notice it with the duplicate uncertainty. We added this explanation L156-159.

10) "Line 228-239: These discussions are not necessary because the Rayleigh process requires sampling from the residues or products of the same reservoir during evolution and the black crust gypsum is not."

Here, we modeled black crusts as the cumulated product, not as instantaneous product. The initial isotopic composition is the same for the whole Parisian basin, and black crusts from different places represent the cumulated products at different residual fraction F, which is the residual sulfates in water that leached the host-rock. Our gypsum precipitation model is now better specified L254-259.

11) " Line 240-252: I think this is because Harris et al (2012)'s fractionation factors are not for multiple steps with multiple oxygen sources and are only applicable in their particular experimental settings."

We agree that they are applicable in their particular experimental settings but they are

the only existing parameters so far, we cannot therefore neglect them. We choose to keep our model in the manuscript L260-270.

12) "Line 256, 257, 261 ...: significant digits should reflect experimental error, in the case of 34S, it should be at most at the second decimal points."

We modified the digits of all d34S and $\triangle$33S (second decimal points).

13) "Line 393-402: This O3-H2O2 proportion exercise is not only too simplified but also invalid because you did not consider the contribution of Fe-Mn catalyzed oxidation by O2 in aqueous condition, a pathway that is known to be significant."

In the new version, we removed this calculation because the main point was only to highlight the required presence of atmospheric sulfates in black crusts ($\triangle$17O > 0.65‰.

14) "Line 444-445: The sentence "resulting in negative $\triangle$33S-$\triangle$36S but not low enough to explain $\triangle$33S < -0.2‰Ź" is ambiguous here."

We wrote it in another way L394-397, specifying the model on figure 6 "resulting in slightly negative $\triangle$33S-$\triangle$36S that could not explain the $\triangle$33S as low as -0.34 ‰ measured in black crusts (yellow frames on Fig. 6)".

15) "Line 521-523: I'd rather see this "microbial ..." sentence deleted entirely."

As we do not know the reaction leading to MIE in black crusts, we think it is important to suggest some possibilities that rely on previous observations. We did not accordingly delete this sentence.

---

## Author Comment (AC2) · 18 Feb 2020

First of all, there was no rebuttal to our interpretations and the comments raised by the second referee were important to improved our manuscript. We almost followed all of them.

1) "The manuscript is too long and there is too much information. I think the most important finding of this manuscript is the observation of negative D33S values. I believe that the authors agree with me as they also put this information in the title. However,

the authors used 5-6 pages to discuss d18O, d34S, and D17O data, which may be distracting. I agree that some discussions in those parts are important to support the authors' interpretation, but in my point of view, many of them may be unnecessary and even irrelevant (e.g., lines 370-375). I understand that everyone has his/her own writing style, but I am afraid that some readers will get lost in this manuscript. My suggestion is that the authors should shorten the d18O, d34S, and D17O discussions, and highlight their most important findings (i.e., the negative D33S data). Though I will leave it for the authors to decide as to whether they prefer keeping their writing style."

We think that before arguing that the origin of the $\Delta$33S anomaly lies in a "process" rather than a "source", it is a pre-requisite to address if S and O isotopic compositions are affected by processes (e.g. SO2 partial oxidation, gypsum precipitation...) even if it only slightly overprints the source signatures. It would otherwise confuse the reader having two sections, one dealing with mixing (d34S-d18O-$\Delta$17O), the other dealing with processes (d34S-$\Delta$33S-$\Delta$36S). We added our developed reasoning L247-251. Given that d34S-d18O variation is explained by a mixing, representing source signatures and that $\Delta$17O-values are evidence for atmospheric aerosols, we know that $\Delta$33S signature is not overprinted and that the source or process leading to this anomaly is atmospheric (not coming from the host-rock). We merged d34S-18O-$\Delta$17O sections, keeping the discussion about processes even if mixing is the main mechanism and reduced the text by $\sim$ 1.5 pages for this part of the discussion, almost 20% of the 50% asked by the first referee. Furthermore, we added details asked by both referees which does not allow us to reduce the text as much. The discussion is very detailed and as the referee said, it is the writing style of our laboratory because we think that most of the time, the interpretation is limited to a mixing without discussing other processes. Furthermore, various scientific communities could be interested in this paper without understanding all concepts covered (e.g. black crusts, stable isotope geochemistry in the atmosphere, mass-independent fractionation). Consequently, we wish to keep a detailed, rigorous and accordingly developed discussion to allow everyone a good comprehension.

2) "In the introduction, the current state and gap of our knowledge on S-MIF (especially D33S) is not well presented. It is true that most aerosol measurements displayed positive D33S values, but negative D33S values were also noted by previous works by Lee et al. (2002), Shaheen et al. (2014), Han et al. (2017), and Lin et al. (2018a). In addition, I suggested the authors highlighted their recent work (Au Yang et al., 2019) in the introduction. Au Yang et al., (2019) suspected that photooxidation of SO2 on the surface of mineral dusts may produce large S-MIF signatures, and the gypsum layer on carbonate studied in this study is a natural laboratory to test their hypothesis and concept model."

We agree and referred to previous works (L61-76) reporting negative $\Delta$33S-values. As suggested by the referee, we also highlighted the study of Au Yang et al., (2019) which proposes a model to counterbalance $\Delta$33S > 0‰ in sulfate aerosols.

3) "The authors listed some end-members to quantify the anthropogenic contribution, but it is not clear how the authors defined their isotopic values. For example, in line 289, it was mentioned that the end-member CAS/PL has a D17O value of zero per mil. Did the author measure the CAS/PL? Is this value just a simple predication? In line 291, it was mentioned that the other end-member possesses d18O values ranging from 5 to 15 per mil. I am not sure where the authors obtained these numbers. Can the authors clarify?"

First, end-members were chosen graphically to encompass all our data on Fig. 4 and 5. This is the reason why the CAS/Pl end-member (which is inferred here) has a $\Delta$17O of 0‰ in agreement with marine sulfates $\Delta$17O. Then, we looked for sources in the literature that match those end-members. The way we chose our end-members is clarified in the text. Concerning the d18O, the 34S-depleted end-member (anthropogenic sulfur) has a variable 18O from approximately 5 to 15‰ to encompass all our black crusts graphically. We chose to limit this mixing calculation to two end-members (explaining the large d18O range for An end-member) because the variation of oxygen isotopes (d18O and $\Delta$17O) reflects various oxidation pathways.

"The definition of anthropogenic emission endmember (d34S = -3 per mil) is also unclear and may be problematic. It is well-known that d34S values of anthropogenic emitted sulfur are highly variable. This fact is also noted by the authors in lines 314-316. Given that the 34S value could be from -30 per mil to 30 per mil as cited by the authors, I am confused why the value of -3 per mil was selected. I checked Montana et al. (2012) cited by the authors but cannot find the value of -3 per mil."

Anthropogenic sulfur, with its wide range of d34S from -30 to 30‰ can match both end-members but the sulfate aerosols in Paris measured by David Au Yang during his PhD show a narrow range between -0.57 and 11.33‰ (L333-334) and the vicinity of the plaster and occurrence of CAS with 34S-enriched sulfates cannot be neglected as potential sources. Therefore, we decided to model a mixing between two different sources, 34S-enriched and depleted. We took an An d34S end-member of -3‰ because it enables to encompass all our data graphically, it is the same as reported by Montana et al. (2008) (and not Montana et al., (2012), sorry for this mistake) and close to the most negative value measured in Paris (d34S = -0.57‰. To highlight the sensitivity of our calculation to the somewhat assumed d34S of -3‰ we also calculated mixing proportions with an An d34S end-member of -10‰ encompassing the black crusts of Antwerp supposed to trap higher amounts of anthropogenic sulfur (Torfs et al., 1997; L350-352).

4) "I would like to point out a possibility that pyrite could be oxidized by H2O2 or even O3. This is shown by a recent talk in the AGU fall meeting this month(https://agu.confex.com/agu/fm19/meetingapp.cgi/Paper/597703). I am not saying that such oxidation pathways play a dominant role in the authors' samples, but the authors should carefully consider this possibility. One can also argue that the authors' samples are mixed by three end-members (CAS/PL, anthropogenic sulfur, and oxidized pyrites)."

We added this information in the manuscript L325-330. Pyrite oxidation by O3 could explain $\Delta 17O > 0‰$ for the 34S-depleted end-member but we argued that so far the

fluxes of common atmospheric constituents for pyrite oxidation, i.e. O2 and H2O, are still higher than O3 amounts.

5) "The D17O calculation in line 397 is wrong. The measured D17O value is also controlled by other oxidants such as OH radical and O2. In the calculation done by the authors, the estimated contribution of S(IV)+O3 oxidation pathway is just the lower limit as the authors assume the S(IV)+H2O2 reaction is the only other oxidation pathway. The authors can also estimate the maximum contribution of S(IV)+O3 oxidation pathway by assuming no contribution from H2O2."

In the new version, we removed this calculation because the main point was only to highlight the presence of atmospheric sulfates in black crusts ($\Delta 17O > 0.65$‰.

6) "Line 514: This is a misleading statement. It is an "interpretation" instead of "observation". Non-zero D33S value is still a prediction and has not been confirmed by any measurement yet. It is important to do such measurements in the future to test the authors' prediction though."

This misleading is corrected in the new version L463-466: "SO2 in the Paris basin still has to be measured to confirm this assumption but so far, this could be consistent with the interpretation that non-zero $\Delta 33S$-values of residual/background atmospheric SO2 are erased by anthropogenic SO2 having zero $\Delta 33S$-values (Au Yang et al., 2019) moving towards the local source(s) of anthropogenic SO2."

7) "In the discussion of magnetic isotope effects, the authors suggested that they cannot rule out the effect of micro-organisms. If isotopic compositions in black crusts were linked to magnetic isotope effects from microbial activities, how do the authors explain the D17O data?"

We think that oxygen isotope compositions are not affected by MIE because we should otherwise expect at least that the most 33S-depleted sample corresponds to the most 17O-depleted-values, which is not the case (added L475-476). Furthermore, sulfates

from black crusts sampled in Paris have Δ17O similar to sulfate aerosols also gathered in Paris by Lee et al., (2002), suggesting that oxygen isotopes record an atmospheric process, not affected by MIE on the surface of building stones.

8) "I suggest putting all legends on all figures. It is difficult to check the caption one by one."

We put all legends on all figures as it was suggested.
* * *

---

## Author Response (AR1)

**Reply to comments of the first referee**

First of all, there was no rebuttal to our interpretations and the comments raised by the first referee were important to improved our manuscript. We almost followed all of them.

1) *Some of the discussion parts are unnecessarily lengthy especially considering the insignificance of the problems in question. The proportion of natural vs. anthropogenic estimates have been done before and usually bear a large uncertainty and is not a critical problem. Those lengthy discussions and estimates are diluting the important discoveries in this study. I suggest trying to trim the text down to 50% of the current length. Focus on new things, the $\Delta^{33}S$, and the $\Delta^{33}S$ and $\Delta^{36}S$ correlation.*

We think that before addressing if $\Delta^{33}S$ anomaly lies in a "process" rather than a "source", it is important to check if S and O isotopic compositions are affected by processes (e.g. $SO_2$ partial oxidation, gypsum precipitation…) even if it only slightly overprint the source signatures. It would otherwise confuse the reader having two sections, one dealing with mixing ($\delta^{34}S$-$\delta^{18}O$-$\Delta^{17}O$), the other dealing with processes ($\delta^{34}S$-$\Delta^{33}S$-$\Delta^{36}S$). We added our developed reasoning L277-292. Given that $\delta^{34}S$-$\delta^{18}O$ variation is explained by a mixing, representing source signatures and that $\Delta^{17}O$-values are evidence for atmospheric aerosols, we know that $\Delta^{33}S$ signature is not overprinted and that the source or process leading to this anomaly is atmospheric (not coming from the host-rock). We merged $\delta^{34}S$-$\delta^{18}O$-$\Delta^{17}O$ sections, keeping the discussion about processes even if mixing is the main mechanism and reduced the text by ~ 1.5 pages for this part of the discussion, almost 20% of the 50% asked by the first referee, whereas the second referee leaves us the decision to reduce or not. Furthermore, we added details asked by both referees which does not allow us to reduce the text as much. The discussion is very detailed because we think that most of the time, the interpretation is limited to a mixing without discussing other processes. Furthermore, various scientific communities could be interested in this paper without understanding all concepts covered (e.g. black crusts, stable isotope geochemistry in the atmosphere, mass-independent fractionation). Consequently, we wish to keep a detailed, rigorous and accordingly developed discussion to allow everyone a good comprehension.

2) *I could guess from the text that at least two writers were writing this manuscript. Make sure the English and flow are consistent.*

We agree and tried to homogenize the English in the manuscript.

3) We took into account all the syntactic comments suggested by the referee, not listed here.

4) *Line 26-29: This is inconsistent with the many published negative $\Delta^{33}S$ data from Beijing, e.g. Han et al., 2017.*

   We clarified this aspect, adding L27 that "except for a few samples", sulfate aerosols have mostly positive $\Delta^{33}S$.

5) For the following comments:

   "Not necessarily going through a $H_2SO_4$ phase" we added "can" L32.

   "Influent" is replaced by "efficient particles" L37.

   "Not necessarily distant" we added "a local or more distant source" L45.

   For the oxidants, we used the simplification suggested by the referee that is the three

   main pathways L44-47.

6) *Line 81: "Intrinsic" is a poor choice here.* and *Line 225: "extrinsic" and "intrinsic" are not ideal words here.*

   We agree but we only reported this term because used in the literature by Kramar et al., (2011) and Kloppmann et al., (2011). Because many distinctions exist for sulfate aerosols (primary/secondary; natural/anthropogenic…), we removed the intrinsic/extrinsic distinction in the discussion, only recalling its use in the literature L100-101.

7) *Line 88-89: Few sulfate samples have been measured for all the 4 sulfur and 3 oxygen isotope compositions together. Thus, this is not a significant thing to say.*

   We wanted to highlight that this is the first time for black crusts samples. We agree it is not the case for sulfate aerosols and we removed this emphasis L107.

8) *Line 105-114: There are numerous conceptual misunderstanding and inaccuracies in these writings. I suggest delete them all. Line 114-115: Some of the deviations maybe still be massdependent under this definition per se. Line 117: I suggest you use 0.5305 for the sake of internal consistency. Note that both 0.515 and 1.889 are the high-temperature limit values for quadruple sulfur isotope system. For triple oxygen isotope system at high-T limit, the exponent value is 0.5305.*

   We explained in another way, L23-128, the concept of mass-dependent vs mass-independent fractionation depending on how large is the deviation from the mass-dependent curve, without using the term "anomaly" which was indeed ambiguous. We also applied 0.5305 for $^{17}\beta$ (L119, L123 and Eq. 1), specifying that all the $\beta$ used here are the high temperature limit (L123), as suggested by the referee.

9) *Line 158: We had a similar correction factor. This correction will have +/-2‰ error (1 sigma). Sample impurity and therefore $O_2$ yield has been the major source of errors. Thus, the actual error for $^{18}O$ of sulfate could be much larger.*

We believe that our correction factor can be applied to all samples without such a significant error because 1) all samples were purified on an ion-exchange resin to get only the sulfates (protocol implemented by Legendre et al., 2017) and 2) the reproducibility between duplicate samples is ~ 0.3‰, lower than the NBS127 reproducibility. If the $O_2$ yield was different for one sample, we should notice it with the duplicate uncertainty. We added this explanation L189-191.

10) *Line 228-239: These discussions are not necessary because the Rayleigh process requires sampling from the residues or products of the same reservoir during evolution and the black crust gypsum is not.*

Here, we modeled black crusts as the cumulated product, not as instantaneous product. The initial isotopic composition is the same for the whole Parisian basin, and black crusts from different places represent the cumulated products at different residual fraction F, which is the residual sulfates in water that leached the host-rock. Our gypsum precipitation model is now better specified L295-300.

11) *Line 240-252: I think this is because Harris et al (2012)'s fractionation factors are not for multiple steps with multiple oxygen sources and are only applicable in their particular experimental settings.*

We agree that they are applicable in their particular experimental settings but they are the only existing parameters so far, we cannot therefore neglect them. We choose to keep our model in the manuscript L302-307.

12) *Line 256, 257, 261 ...: significant digits should reflect experimental error, in the case of $\delta^{34}S$, it should be at most at the second decimal points.*

We modified the digits of all $\delta^{34}S$ and $\Delta^{33}S$ (second decimal points).

13) *Line 393-402: This $O_3$-$H_2O_2$ proportion exercise is not only too simplified but also invalid because you did not consider the contribution of Fe-Mn catalyzed oxidation by $O_2$ in aqueous condition, a pathway that is known to be significant.*

In the new version, we removed this calculation because the main point was only to highlight the required presence of atmospheric sulfates in black crusts ($\Delta^{17}O > 0.65‰$).

*14) Line 444-445: The sentence "resulting in negative Δ33S-Δ36S but not low enough to explain Δ33S < -0.2‰˙'' is ambiguous here.*

We wrote it in another way L632-633, specifying the model on figure 6 "resulting in slightly negative $\Delta^{33}S$-$\Delta^{36}S$ that could not explain the $\Delta^{33}S$ as low as -0.34 ‰ measured in black crusts (yellow frames on Fig. 6)".

*15) Line 521-523: I'd rather see this "microbial ..." sentence deleted entirely.*

As we do not know the reaction leading to MIE in black crusts, we think it is important to suggest some possibilities that rely on previous observations. We did not accordingly delete this sentence.

**Reply to comments of Mang Lin, second referee**

First of all, there was no rebuttal to our interpretations and the comments raised by the second referee were important to improved our manuscript. We almost followed all of them.

*1) The manuscript is too long and there is too much information. I think the most important finding of this manuscript is the observation of negative D33S values. I believe that the authors agree with me as they also put this information in the title. However, the authors used 5-6 pages to discuss d18O, d34S, and D17O data, which may be distracting. I agree that some discussions in those parts are important to support the authors' interpretation, but in my point of view, many of them may be unnecessary and even irrelevant (e.g., lines 370-375). I understand that everyone has his/her own writing style, but I am afraid that some readers will get lost in this manuscript. My suggestion is that the authors should shorten the d18O, d34S, and D17O discussions, and highlight their most important findings (i.e., the negative D33S data). Though I will leave it for the authors to decide as to whether they prefer keeping their writing style.*

We think that before arguing that the origin of the $\Delta^{33}S$ anomaly lies in a « process » rather than a « source », it is a pre-requisite to address if S and O isotopic compositions are affected by processes (e.g. $SO_2$ partial oxidation, gypsum precipitation…) even if it only slightly overprints the source signatures. It would otherwise confuse the reader having two sections, one dealing with mixing ($\delta^{34}S$-$\delta^{18}O$-$\Delta^{17}O$), the other dealing with processes ($\delta^{34}S$-$\Delta^{33}S$-$\Delta^{36}S$). We added our developed reasoning L277-292. Given that $\delta^{34}S$-$\delta^{18}O$ variation is explained by a mixing, representing source signatures and that $\Delta^{17}O$-values are evidence for atmospheric aerosols, we know that $\Delta^{33}S$ signature is not overprinted and that the source or process leading to this anomaly is atmospheric (not coming from the host-rock). We merged $\delta^{34}S$-$\delta^{18}O$-$\Delta^{17}O$ sections, keeping the discussion

about processes even if mixing is the main mechanism and reduced the text by ~ 1.5 pages for this part of the discussion, almost 20% of the 50% asked by the first referee. Furthermore, we added details asked by both referees which does not allow us to reduce the text as much. The discussion is very detailed and as the referee said, it is the writing style of our laboratory because we think that most of the time, the interpretation is limited to a mixing without discussing other processes. Furthermore, various scientific communities could be interested in this paper without understanding all concepts covered (e.g. black crusts, stable isotope geochemistry in the atmosphere, mass-independent fractionation). Consequently, we wish to keep a detailed, rigorous and accordingly developed discussion to allow everyone a good comprehension.

2) *In the introduction, the current state and gap of our knowledge on S-MIF (especially D33S) is not well presented. It is true that most aerosol measurements displayed positive D33S values, but negative D33S values were also noted by previous works by Lee et al. (2002), Shaheen et al. (2014), Han et al. (2017), and Lin et al. (2018a). In addition, I suggested the authors highlighted their recent work (Au Yang et al., 2019) in the introduction. Au Yang et al., (2019) suspected that photooxidation of SO2 on the surface of mineral dusts may produce large S-MIF signatures, and the gypsum layer on carbonate studied in this study is a natural laboratory to test their hypothesis and concept model.*

We agree and referred to previous works (L60-75) reporting negative $\Delta^{33}$S-values. As suggested by the referee, we also highlighted the study of Au Yang et al., (2019) which proposes a model to counterbalance $\Delta^{33}$S > 0‰ in sulfate aerosols.

3) *The authors listed some end-members to quantify the anthropogenic contribution, but it is not clear how the authors defined their isotopic values. For example, in line 289, it was mentioned that the end-member CAS/PL has a D17O value of zero per mil. Did the author measure the CAS/PL? Is this value just a simple predication? In line 291, it was mentioned that the other end-member possesses d18O values ranging from 5 to 15 per mil. I am not sure where the authors obtained these numbers. Can the authors clarify?*

First, end-members were chosen graphically to encompass all our data on Fig. 4 and 5. This is the reason why the CAS/Pl end-member (which is inferred here) has a $\Delta^{17}$O of 0‰, in agreement with marine sulfates $\Delta^{17}$O. Then, we looked for sources in the literature that match those end-members. The way we chose our end-members is clarified in the text. Concerning the $\delta^{18}$O, the $^{34}$S-depleted end-member (anthropogenic sulfur) has a variable $\delta^{18}$O from approximately 5 to 15‰ to encompass all our black crusts graphically. We chose to limit this mixing calculation to two end-members (explaining the large $\delta^{18}$O

range for An end-member) because the variation of oxygen isotopes ($\delta^{18}O$ and $\Delta^{17}O$) reflects various oxidation pathways.

*The definition of anthropogenic emission endmember (d34S = -3 per mil) is also unclear and may be problematic. It is well-known that d34S values of anthropogenic emitted sulfur are highly variable. This fact is also noted by the authors in lines 314-316. Given that the 34S value could be from -30 per mil to 30 per mil as cited by the authors, I am confused why the value of -3 per mil was selected. I checked Montana et al. (2012) cited by the authors but cannot find the value of -3 per mil.*

Anthropogenic sulfur, with its wide range of $\delta^{34}S$ from -30 to 30‰ can match both end-members but the sulfate aerosols in Paris measured by David Au Yang during his PhD show a narrow range between -0.57 and 11.33‰ (L399-401) and the vicinity of the plaster and occurrence of CAS with $^{34}S$-enriched sulfates cannot be neglected as potential sources. Therefore, we decided to model a mixing between two different sources, $^{34}S$-enriched and depleted. We took an An $\delta^{34}S$ end-member of -3‰ because it enables to encompass all our data graphically, it is the same as reported by Montana et al. (2008) (and not Montana et al., (2012), sorry for this mistake) and close to the most negative value measured in Paris ($\delta^{34}S$ = -0.57‰). To highlight the sensitivity of our calculation to the somewhat assumed $\delta^{34}S$ of -3‰, we also calculated mixing proportions with an An $\delta^{34}S$ end-member of -10‰, encompassing the black crusts of Antwerp supposed to trap higher amounts of anthropogenic sulfur (Torfs et al., 1997; L417).

4) *I would like to point out a possibility that pyrite could be oxidized by H2O2 or even O3. This is shown by a recent talk in the AGU fall meeting this month(https://agu.confex.com/agu/fm19/meetingapp.cgi/Paper/597703). I am not saying that such oxidation pathways play a dominant role in the authors' samples, but the authors should carefully consider this possibility. One can also argue that the authors' samples are mixed by three end-members (CAS/PL, anthropogenic sulfur, and oxidized pyrites).*

We added this information in the manuscript L371-396. Pyrite oxidation by $O_3$ could explain $\Delta^{17}O > 0$‰ for the $^{34}S$-depleted end-member but we argued that so far the fluxes of common atmospheric constituents for pyrite oxidation, i.e. $O_2$ and $H_2O$, are still higher than $O_3$ amounts.

5) *The D17O calculation in line 397 is wrong. The measured D17O value is also controlled by other oxidants such as OH radical and O2. In the calculation done by the authors, the*

*estimated contribution of S(IV)+O3 oxidation pathway is just the lower limit as the authors assume the S(IV)+H2O2 reaction is the only other oxidation pathway. The authors can also estimate the maximum contribution of S(IV)+O3 oxidation pathway by assuming no contribution from H2O2.*

In the new version, we removed this calculation because the main point was only to highlight the presence of atmospheric sulfates in black crusts ($\Delta^{17}O > 0.65‰$).

6) *Line 514: This is a misleading statement. It is an "interpretation" instead of "observation". Non-zero D33S value is still a prediction and has not been confirmed by any measurement yet. It is important to do such measurements in the future to test the authors' prediction though.*

This misleading is corrected in the new version L707-710: "$SO_2$ in the Paris basin still has to be measured to confirm this assumption but so far, this could be consistent with the interpretation that non-zero $\Delta^{33}S$-values of residual/background atmospheric $SO_2$ are erased by anthropogenic $SO_2$ having zero $\Delta^{33}S$-values (Au Yang et al., 2019) moving towards the local source(s) of anthropogenic $SO_2$."

7) *In the discussion of magnetic isotope effects, the authors suggested that they cannot rule out the effect of micro-organisms. If isotopic compositions in black crusts were linked to magnetic isotope effects from microbial activities, how do the authors explain the D17O data?*

We think that oxygen isotope compositions are not affected by MIE because we should otherwise expect at least that the most $^{33}S$-depleted sample corresponds to the most $^{17}O$-depleted-values, which is not the case (added L716-720). Furthermore, sulfates from black crusts sampled in Paris have $\Delta^{17}O$ similar to sulfate aerosols also gathered in Paris by Lee et al., (2002), suggesting that oxygen isotopes record an atmospheric process, not affected by MIE on the surface of building stones.

8) *I suggest putting all legends on all figures. It is difficult to check the caption one by one.*

We put all legends on all figures as it was suggested.

[revised manuscript text omitted]

| | | | | | |
|---|---|---|---|---|---|
| ME77-2 | 48° 32' 20'' N
2° 39' 33 '' E | 22° N | 210 | 1.3 | Not directly exposed |
| AV77-1 | 48° 24' 15 '' N
2° 43' 2'' E | 73° N | 230 | 1.3 – 2.3 | Directly exposed |
| SE89-1 | 48° 12' 8'' N
3° 16' 24'' E | 263° N | 270 | 1.7 | Not directly exposed |
| PY89-1 | 48° 17' 16'' N
3° 12' 16'' E | 343° N | 263 | 1.5 – 2.1 | Not directly exposed |
| PO77-1 | 48° 33' 38'' N
3° 17' 29'' E | 23° N | 230 | 1.5 – 2.0 | Not directly exposed |
| NA77-1 | 48° 33' 26'' N
3° 0' 25'' E | 351° N | 230 | 2.5 | Not directly exposed |

230 **Table 1** Characteristics of black crusts samples. Their name was given according to the city and the department where they are located and following the number of samples gathered at the same place (NA77-1: NA = Nangis; 77 = department; -1 = first sample collected).

235

240

245

250

255

| (2σ) | $\delta^{18}O$ ± 0.5 ‰ | $\Delta^{17}O$ ± 0.05 ‰ | $\delta^{34}S$ ± 0.20 ‰ | $\Delta^{33}S$ ± 0.01 ‰ | $\Delta^{36}S$ ± 0.20 ‰ | Distance from coastline (km) |
|---|---|---|---|---|---|---|
| PO78-2 | 9.7 | 0.27 | 1.34 | -0.17 | -0.68 | 150 |
| AV77-1 | 9.8 | 1.03 | 2.25 | -0.07 | -0.33 | 230 |
| PY89-1 | 9.8 | 0.84 | 0.46 | -0.21 | -0.47 | 263 |
| PO77-1 | 12.4 | 2.56 | -2.66 | -0.07 | -0.50 | 231 |
| BG76-1 | 13.1 | 1.65 | 4.94 | -0.10 | -0.47 | 21 |
| MV95-1 | 15.5 | 0.46 | 10.17 | -0.04 | -0.40 | 110 |
| BR91-1 | 14.8 | 0.18 | 13.51 | -0.02 | -0.22 | 190 |
| JU76-1 | 9.9 | 0.78 | 2.93 | -0.01 | -0.24 | 50 |
| TV27-1 | 16.7 | 0.27 | 13.99 | -0.02 | -0.54 | 95 |
| YV76-1 | 12.9 | 1.01 | 10.51 | -0.04 | -0.27 | 28 |
| DR28-1 | 7.5 | 0.70 | -2.22 | -0.30 | -0.34 | 115 |
| FE76-1 | 10.3 | 1.35 | 0.95 | -0.16 | -0.59 | 0.5 |
| NO27-1 | 14.2 | 1.64 | 1.78 | -0.04 | -0.45 | 103 |
| ME77-2 | 10.5 | 1.07 | -0.54 | -0.21 | -0.29 | 210 |
| PA13-2 | 7.9 | 0.81 | -0.87 | -0.10 | -0.58 | 170 |
| MLJ78-1 | 15.2 | 0.36 | 8.30 | 0.00 | -0.52 | 135 |
| EV27-1 | 10.4 | 0.79 | 6.60 | -0.05 | -0.37 | 84 |
| FE76-2 | 8.7 | 1.19 | -1.15 | -0.04 | -0.76 | 1.1 |
| SE89-1 | 10.4 | 0.81 | 3.15 | -0.11 | -0.41 | 270 |
| BU76-2 | 10.3 | 1.43 | -0.11 | -0.11 | -0.50 | 45 |
| MR27-1 | 11.2 | 0.84 | 2.82 | -0.15 | -0.49 | 28 |
| PA14-1 | 9.2 | 0.17 | 1.55 | -0.21 | -0.56 | 171 |
| TO77-1 | 13.3 | 0.08 | 12.03 | -0.02 | -0.64 | 202 |
| NA77-1 | 9.8 | 1.42 | 2.40 | -0.11 | -0.49 | 232 |
| PA5-1 | 10.5 | 0.89 | 0.47 | -0.34 | -0.32 | 172 |
| SM94-1 | 10.4 | 0.43 | 3.60 | -0.13 | -0.28 | 175 |
| CC76-1 | 9.6 | 0.25 | 5.82 | -0.07 | -0.43 | 37 |

**Table 2** $\delta^{18}O$, $\delta^{34}S$, $\Delta^{17}O$, $\Delta^{33}S$ and $\Delta^{36}S$ measures of each sample with the distance from coastline.

| Page 3 : [1] Supprimé | Utilisateur de Microsoft Office | 21/02/2020 15:48:00 |

Thus, black crusts were never investigated for all the oxygen ($\delta^{18}O$, $\delta^{17}O$ and therefore $\Delta^{17}O$) and sulfur ($\delta^{34}S$, $\delta^{33}S$, $\delta^{36}S$ and therefore $\Delta^{33}S$ and $\Delta^{36}S$) isotopic ratios contained in sulfate and more specifically, in quantifying the different oxidation channels involved in the sulfate aerosols formation in the troposphere.

| Page 9 : [2] Supprimé | Utilisateur de Microsoft Office | 21/02/2020 15:48:00 |

Montana et al. (2012). Using a mass balance calculation, CAS/plaster proportions range between 2 and 81 % with an average ~ 30 %. Therefore, the host rock sulfate is on average not the main S-provider, highlighting atmospheric sulfate aerosols sampling by black crusts. In order to go further in the anthropogenic sources (primary and/or secondary sulfate aerosols) and oxidation pathways characterization, we combined, in the following section, the measured $\Delta^{17}O$ to $\delta^{34}S$-$\delta^{18}O$ systematic.

**The $\Delta^{17}O$ values as a proxy for $SO_2$ oxidation pathways**

Figure 5 shows samples having near zero $\Delta^{17}O$-values with $\delta^{34}S$-values from -3 up to 14 ‰ and $\Delta^{17}O$-values up to 2.6 ‰, with $\delta^{34}S$-values that are < 10 ‰ when $\Delta^{17}O$ > 1 ‰. These values are consistent with $\Delta^{17}O$-values of sulfate aerosols worldwide, that range from 0.14 to 3 ‰ with a mean ~ 0.78 ‰, collected in rainwater or on filters in La Jolla (USA), Baton Rouge (USA), Bakerfield (USA), White Mountain Research Station (WMRS, USA), Wuhan (China) (Bao et al., 2001a; Jenkins and Bao, 2006; Lee and Thiemens, 2001; Li et al., 2013; Romero and Thiemens, 2003). The lack of correlation between $\Delta^{17}O$ and the distance from coastline (Fig. S1b) is consistent with the evidence that $\delta^{34}S$-values are also not correlated with distance from coastline, confirming that DMS, which is mostly oxidized by ozone and carries high $\Delta^{17}O$-values (Alexander et al., 2012), is less significant compared to anthropogenic sulfur.

Large positive $\Delta^{17}O$ anomalies in sulfate aerosols are inherited from their atmospheric oxidants, that were ultimately produced during $O_3$-photochemically induced genesis. In theory, other mechanisms exist such as magnetic isotope effect (see section 5.3.2) but have not been recognized yet. Sulfur dioxide will undergo oxidation via OH radical (in gaseous phase) or $H_2O_2$, $O_3$ or $O_2$ catalyzed by TMI (in aqueous phase) as well as other potential oxidants such as $NO_2$ or Criegee radicals, in the troposphere. As $NO_2$ and Criegee radicals are minor species, these have accordingly been less studied with respect to $\Delta^{17}O$ and are generally omitted. Resulting from photochemical reactions, $O_3$ molecules possess oxygen

| Page 9 : [3] Déplacé vers la page 6 (Déplacement n°1) | Utilisateur de Microsoft Office 21/02/2020 15:48:00 |

-MIF compositions with $\Delta^{17}O$ ~ 35 ‰ (Janssen et al., 1999; Lyons, 2001; Mauersberger et al., 1999)

| Page 9 : [4] Déplacé vers la page 6 (Déplacement n°2) | Utilisateur de Microsoft Office 21/02/2020 15:48:00 |

-TMI have mass-dependent composition with $\Delta^{17}O \sim 0$ (Dubey et al., 1997; Holt et al., 1981; Lyons, 2001) and $\sim -0.34$ ‰ (Barkan and Luz, 2005) respectively. Savarino (2000) measured the O-isotopic compositions of sulfates derived from these

| Page 9 : [5] Supprimé | Utilisateur de Microsoft Office | 21/02/2020 15:48:00 |

different oxidation pathways and showed that the OH and $O_2$-TMI oxidation channels do not result in a mass-independent fractionation ($\Delta^{17}O = 0$ and $-0.09$ ‰ respectively) whereas $O_3$ and $H_2O_2$ radicals transfer ¼ and ½ respectively of their isotopic anomaly to the sulfate thus resulting in a mass-independent fractionation ($\Delta^{17}O = 8.75$ ‰ and $0.65$ ‰ respectively) (e.g.

| Page 9 : [6] Déplacé vers la page 6 (Déplacement n°3)     Utilisateur de Microsoft Office |
| 21/02/2020 15:48:00 |

Bao et al., 2001a; Bao et al., 2000; Bao et al., 2001b; Bao et al., 2010; Jenkins and Bao, 2006; Lee et al., 2002; Lee and Thiemens, 2001; Li et al., 2013; Martin et al., 2014). Mass-dependent isotopic fractionation during $SO_2$ oxidation may change $\delta^{17}O$ and $\delta^{18}O$ but not the $\Delta^{17}O$ that only depends on the mixing of O-reservoirs with variable $\Delta^{17}O$.

| Page 9 : [7] Supprimé | Utilisateur de Microsoft Office | 21/02/2020 15:48:00 |

Therefore, samples with low $\delta^{34}S$-values, consisting mostly of anthropogenic sulfur (section 5.1), and $\Delta^{17}O > 0.65$ ‰ obviously point to a significant anthropogenic $SO_2$ fraction oxidized by $O_3 + H_2O_2$ or by $O_3$ and to a lesser extent by $O_2$

| Page 9 : [8] Supprimé | Utilisateur de Microsoft Office | 21/02/2020 15:48:00 |

and correspond to samples with significant atmospheric sulfate aerosols. As the combustion does not produce sulfates with mass-independent signatures (Lee et al., 2002), primary sulfate aerosols have $\Delta^{17}O \sim 0$ ‰, sea-salt sulfates as well, thus samples with $\Delta^{17}O < 0.65$ ‰ and low $\delta^{34}S$-values then either represent primary anthropogenic sulfate aerosols and/or $SO_2$ oxidized by OH or $O_2$-TMI and/or a subtle mixing of oxidants to yield near-zero $\Delta^{17}O$ (Fig.

| Page 9 : [9] Supprimé | Utilisateur de Microsoft Office | 21/02/2020 15:48:00 |

Samples having the highest $\delta^{34}S$ values were identified as being representative of host-rock sulfates (CAS/PL end-member) and their $\Delta^{17}O$-values near zero is in agreement with this origin. Indeed, marine sulfates can have $\Delta^{17}O$-values down to $-0.70$ ‰ in the geological record (a consequence of high $pCO_2$) during the Marinoan, $\sim 635$ Myr ago, but most of the time, $\Delta^{17}O$-values are typically around 0 and $> -0.2$ ‰ (Bao et al., 2008).

Our $\Delta^{17}O$-values representing the first data in black crusts, it could be speculated that some unexpected processes such as magnetic isotope effects (see section 5.3.2) occurred during black crust formation, i.e.

| Page 9 : [10] Supprimé | Utilisateur de Microsoft Office | 21/02/2020 15:48:00 |

recalculated here for $f_{MI} + f_{MD} = 1$ where MI and MD denote mass-independent and mass-dependent fractionation respectively) and $\sim 55$ % modeled by

| Page 9 : [11] Supprimé | Utilisateur de Microsoft Office | 21/02/2020 15:48:00 |

. Considering now $\Delta^{17}O_{O3}$ = 8.75 ‰ and $\Delta^{17}O_{H2O2}$ = 0.65 ‰ of sulfates derived from the $SO_2$ oxidation by $O_3$ and $H_2O_2$ respectively, we can calculate the proportions (or fluxes) of the two oxidation channels around 4 ($O_3$) and 96 % ($H_2O_2$), following mass balance Eq. (4) Lee and Thiemens (2001) :

| Page 9 : [12] Supprimé | Utilisateur de Microsoft Office | 21/02/2020 15:48:00 |

$$\Delta^{17}O_{measured} = f(O_3) \text{ x } \Delta^{17}O_{O3} + f(H_2O_2) \text{ x } \Delta^{17}O_{H2O2} \qquad (4)$$

It is noteworthy that the estimated proportion of secondary sulfate in the black crust is highly dependent on the oxidation channel fluxes. For instance, a 5% increase in the $O_3$ oxidation channel flux decreases the secondary sulfate proportion in the black crust by 30%. Therefore, even if precise quantification is not possible, we can assume that secondary sulfate aerosols, mostly formed by oxidation of $SO_2$ by $H_2O_2$, dominate in black crusts and hence result from aqueous phase reaction.

**5.3. Black crusts S-MIF signature**

**5.3**

| Page 9 : [13] Supprimé | Utilisateur de Microsoft Office | 21/02/2020 15:48:00 |

The $\Delta^{17}O$-parameter provides key evidence for $SO_2$ oxidation by "atmospheric" oxidants but does not allow distinction between (1) $SO_2$ oxidized in the atmosphere generating secondary sulfate aerosols or (2) $SO_2$ deposited and oxidized on the building stone. This question can be addressed using $\Delta^{33}S$-$\Delta^{36}S$ systematics.

| Page 12 : [14] Supprimé | Utilisateur de Microsoft Office | 21/02/2020 15:48:00 |

Review of available literature shows that either product or residue

| Page 23 : [15] Supprimé | Utilisateur de Microsoft Office | 21/02/2020 15:48:00 |

[Figure]

**Fig. 4** Evolution of $\delta^{34}S$ in function of $\delta^{18}O$ in black crusts sulfates. Grey points are from this study, green and red points represent the isotopic compositions of black crusts from Antwerp (Torfs et al., 1997) and Venice (Longinelli and Bartelloni, 1978) respectively. Orange and blue points represent the isotopic compositions of black crusts from Ljubljana (Kramar et al., 2011) and Bourges (Vallet et al., 2006) respectively, probing potentially an oxidized pyrite source. The yellow star represents the modern seawater

[Figure]

**Legend:**

- ● Black crust (this study)
- ○ Urban aerosols
- ● Combustion process

Theoretical isotopic compositions of sulfates formed by only one oxidation pathway for 10 °C < T < 50 °C

- OH
- H₂O₂
- O₂-TMI
- NO₂

Theoretical isotopic compositions of sulfates formed following 27 % OH; 18 % O₂-TMI; 55 % H₂O₂ (Sofen et al. 2011)

- T>20°C
- T<20°C

Figure labels within plot: Cumulated BC sulfates; $^{33}\beta = 0.9$, $^{36}\beta = 1.9$; OH; NO₂; Urban aerosols; Cumulated secondary aerosols; $^{33}\beta = 0.515$, $^{36}\beta = 1.9$; O₂-TMI (T < 20°C); H₂O₂; O₂-TMI (T > 20°C).

Axes: $\Delta^{36}S$ (‰) vs $\Delta^{33}S$ (‰).